# Unravelling *Diaporthe* Species Associated with Woody Hosts from Karst Formations (Guizhou) in China

**DOI:** 10.3390/jof6040251

**Published:** 2020-10-27

**Authors:** Asha J. Dissanayake, Ya-Ya Chen, Jian-Kui (Jack) Liu

**Affiliations:** 1Fungal Research Laboratory, School of Life Science and Technology, University of Electronic Science and Technology of China, Chengdu 611731, China; asha.janadaree@yahoo.com; 2Institute of Crop Germplasm Resources, Guizhou Academy of Agricultural Sciences, Guiyang 550006, China; wmlove@163.com; 3Guizhou Key Laboratory of Agricultural Biotechnology, Guizhou Academy of Agricultural Sciences, Guiyang 550006, China

**Keywords:** seven new taxa, asexual morph, *Diaporthaceae*, phylogeny, taxonomy

## Abstract

Though several *Diaporthe* species have been reported in China, little is known about the species associated with nature reserves in Guizhou province. During a survey of fungi in six nature reserves in Guizhou province of China, thirty-one *Diaporthe* isolates were collected from different woody hosts. Based on morphology, culture characteristics and molecular phylogenetic analysis, these isolates were characterized and identified. Phylogenetic analysis of internal transcribed spacer region (ITS), combined with translation elongation factor 1-alpha (*tef*), β-tubulin (*tub*), calmodulin (*cal*) and histone H3 (*his*) gene regions identified five known *Diaporthe* species and seven distinct lineages representing novel *Diaporthe* species. The details of five known species: *Diaporthe cercidis, D. cinnamomi, D. conica, D. nobilis* and *D. sackstonii* are given and the seven new species *D. constrictospora*, *D. ellipsospora*, *D. guttulata*, *D. irregularis*, *D. lenispora*, *D. minima*, and *D. minusculata* are introduced with detailed descriptions and illustrations. This study revealed a high diversity of previously undescribed *Diaporthe* species associated with woody hosts in various nature reserves of Guizhou province, indicating that there is a potential of *Diaporthe* species remains to be discovered in this unique landform (Karst formations) in China. Interestingly, the five known *Diaporthe* species have been reported as pathogens of various hosts, and this could indicate that those newly introduced species in this study could be potentially pathogenic pending further studies to confirm.

## 1. Introduction

*Diaporthe* Nitschke (including the *Phomopsis* asexual morph) belongs to family Diaporthaceae, order Diaporthales and class Sordariomycetes [1,2,3] and its species are found worldwide on a diverse range of host plants as endophytes, pathogens and saprobes [4]. Rossman et al. [5] proposed the name *Diaporthe* over *Phomopsis*, as both names are well known amongst plant pathologists and subsequent studies have adopted the latter generic name [4,6,7,8,9,10]. More than 1100 epithets for *Diaporthe* and 986 for *Phomopsis* are listed in Index Fungorum (2020) (http://www.indexfungorum.org/, accessed August 2020) with names often based on host association. Many *Diaporthe* species that are morphologically similar have proven to be genetically distinct [11,12], and several isolates formerly identified based on their hosts were shown to represent different taxa [1]. *Diaporthe* represents a highly complex genus containing numerous cryptic species. In recent studies, *Diaporthe* species have been distinguished mainly by their molecular phylogenies, and the best five gene regions to conduct a multi-gene phylogenetic analysis are ITS, *tef*, *tub*, *cal* and *his* [4,13,14,15,16,17,18,19].

A nature reserve is a protected area of importance for flora, fauna or landscapes of geological or other special interest, which is reserved and managed for purposes of conservation and to provide special opportunities for study or research [20]. The Karst region of Guizhou province is comprised of abundant nature reserves that provide a wide range of ecosystem services such as water supply, soil fertility, ecotourism, recreation, biodiversity conservation and carbon sequestration [20]. However, there are few scientific evaluations made for fungi in national nature reserves and national forest parks in Guizhou province, China [21,22]. During the investigation carried out in 2017 to 2019, several isolates of *Diaporthe* species were collected from six nature reserves in Guizhou province including Fanjing mountain, Guiyang Huaxi wetland park, Guiyang Xiaochehe wetland park, Maolan nature reserve, Suiyang broad water nature reserve and Xingyi Wanfenglin. Fungi isolated from forest trees in China were recorded in old fungal literature, however, most of them lack living culture and molecular data [23,24]. Although several species of *Diaporthe* have been previously recorded from Guizhou province with details of culture and molecular data [25,26], little is known to associate these with hosts in nature reserves. Thus, the aim of this study is to describe and illustrate *Diaporthe* taxa from nature reserves in the Karst region of Guizhou province based on morphological characters and phylogenies derived from combined ITS, *tef*, *tub*, *cal* and *his* gene sequences.

## 2. Materials and Methods

### 2.1. Isolation of Fungal Material, Morphology and Culture Characteristics

From 2017 to 2019, thirty-one *Diaporthe* specimens were collected in field surveys of decaying saprobic woody hosts in different nature reserves including Fanjing mountain, Guiyang Huaxi wetland park, Guiyang Xiaochehe wetland park, Maolan nature reserve, Suiyang broad water nature reserve and Xingyi Wanfenglin in Karst region of Guizhou province (Table 1). Collected samples were taken to the laboratory for isolation and photographed, documented and then kept at 4 °C for further study.

Species identification was primarily based on morphological observation of the conidiomata or ascomata from host materials and micromorphology supplemented by culture characteristics. Morphological observations were made using a Motic SMZ (Stereoscopic Zoom Microscope) 168 series stereomicroscope and photographed by a Nikon E80i microscope-camera system. Measurements were made with the Tarosoft (R) Image FrameWork [27] and images used for figures were processed with Adobe Photoshop CS v. 5. Single spore isolations were prepared following the method of Chomnunti et al. [28]. Spore germination on 2% water agar (WA) was examined after 24 h and germinating spores were transferred to potato dextrose agar (PDA) media. Cultures were incubated at 25 °C in the dark and colony morphology and conidial characteristics were examined for a total of 31 isolates. Colony color was determined according to Rayner [29] after 5 d to 10 d on PDA at 25 °C in the dark. More than 20 conidiomata/ascomata, 30 asci, and 50 conidia/ascospores were measured to calculate the mean size/length and respective standard deviations (SD). Conidial shape, color and guttulation were also recorded.

Herbarium specimens were deposited at the Herbarium of Cryptogams, Kunming Institute of Botany Academia Sinica (KUN-HKAS), Kunming, China and herbaria of Guizhou Academy of Agricultural Sciences (GZAAS), Guiyang, China. The living cultures were deposited in the China General Microbiological Culture Collection Center (CGMCC) in Beijing, China and Guizhou Culture Collection (GZCC) in Guiyang, China and (Table 1).

### 2.2. Molecular Based Amplification

Fungal mycelium of 7 d old cultures was scraped for the extraction of genomic DNA using Biospin Fungus Genomic DNA Extraction Kit (BioFlux^®^) following the manufacturer’s protocol (Hangzhou, China). For the identification of *Diaporthe* specimens, the internal transcribed spacer region (ITS) was sequenced for all isolates and BLAST search (basic local alignment search tool) at GenBank was used to reveal the closest matching taxa. Besides ITS gene sequence data, translation elongation factor 1-alpha (*tef*), β-tubulin (*tub*), calmodulin (*cal*) and histone H3 (*his*) gene regions were also employed to support the species identification. The ITS region was amplified using universal primers ITS1 and ITS4 [30]. The target region of the *tef* gene was amplified using primer pairs EF-728F and EF-986R [31]. A portion of the *tub* gene was amplified using the primers BT2a and BT2b [32], while the primer pair CAL228F and CAL737R was used to amplify the *cal* gene region [31]. The primers CYLH3F [33] and H3-1b [32] were used to amplify part of the *his* gene. The PCR reactions were accomplished in a Bio Rad C1000 thermal cycler. The amplification procedure was performed in a 50 μL reaction volume containing 5–10 ng DNA, 0.8 units Taq polymerase, 1X PCR buffer, 0.2 mM dNTP, 0.3 μm of each primer with 1.5 mM MgCl_2._ Following the PCR amplification, products were visualized on 1% agarose gel under UV light using a Gel Doc^TM^ XR Molecular Imager following ethidium bromide staining. PCR products were purified using minicolumns, purification resin and buffer according to the manufacturer’s protocols (Amersham product code: 27–9602–01). Sequence analysis was carried out by Shanghai Sangon Biological Engineering Technology and Services Co., Ltd. (Shanghai, China).

### 2.3. Sequence Alignment and Phylogenetic Analyses

To assure the sequence quality, the resulting sequence chromatograms were checked using BioEdit v.5 [34]. An overview phylogenetic tree (provided as a Appendix A) for the genus *Diaporthe* was constructed from ITS sequence data of all type/ex-type/neo-type *Diaporthe* species from previous studies [10,13,14,15,33,35,36,37,38,39,40,41,42,43,44,45]. Considering this ITS tree, another phylogenetic analysis was conducted including all the isolates obtained in this study (Table 1) together with several closely associated *Diaporthe* species (Table 2). *Diaporthella corylina* (CBS 121124) was selected as the outgroup taxon. The sequences were retrieved from GenBank and aligned with the sequences obtained in this study using MAFFT [46] (http://www.ebi.ac.uk/Tools/msa/mafft/) and manually edited with BioEdit [34] for a maximum alignment. Phylogenetic analysis was performed by using PAUP (Phylogenetic Analysis Using Parsimony) v.4.0b10 for maximum parsimony (MP) method [47], RAxML for maximum likelihood (ML) method [48] and MrBayes v.3.1.2 for Bayesian Inference (BI) method [49]. The best model of evolution was determined by MrModeltest v. 2.3 [50]. Maximum likelihood analyses was accomplished using RAxML GUI v. 0.9b2 [51] with 1000 non-parametric bootstrapping iterations, using the general time-reversible model (GTR) with a discrete gamma distribution. The best scoring trees were chosen with final likelihood values. 

Ambiguous regions in the MP alignment were excluded, and gaps were treated as missing data. The stability of the trees was evaluated by 1000 bootstrap replications. Branches of zero length were collapsed, and all multiple parsimonious trees saved. Statistics including tree length (TL), consistency index (CI), retention index (RI), relative consistency index (RC) and homoplasy index (HI) were calculated. Differences between the trees inferred under different optimality criteria were evaluated using Kishino–Hasegawa tests (KHT) [52].

Bayesian analyses were performed in MrBayes v.3.0b4 [49] and posterior probabilities (PP) were determined by Markov Chain Monte Carlo sampling (MCMC). MrModeltest v. 2.3 [50] was used for the statistical selection of the best-fit model of nucleotide substitutions and was integrated into the analysis. Six simultaneous Markov chains were run for 10^6^ generations; sampling the trees at every 100th generation. From the 10,000 trees obtained, the first 2000 representing the burn-in phase were discarded. The remaining 8000 trees were used for calculating posterior probabilities in the majority rule consensus tree.

The details of the fungal strains obtained in this study are listed in Table 1 with information of the type cultures and sequence data. Sequences generated in this study were deposited in GenBank (Table 1); alignments and trees were deposited in TreeBASE (www.treebase.org, study ID S27013). Reviewer access URL: http://purl.org/phylo/treebase/phylows/study/TB2:S27013?x-access-code=1369710211c386567d8b43ba36f49adf&format=html. Taxonomic novelties were submitted to the Faces of Fungi database [53], Index Fungorum (Index Fungorum 2020) and MycoBank (www.mycobank.org) [33].

## 3. Results

### 3.1. Phylogenetic Analyses

Saprobic specimens sampled from numerous woody hosts in six nature reserves in the Karst region of Guizhou province, China resulted in the isolation of thirty-one isolates of *Diaporthe* (Table 1, Figure 1). The ITS gene was employed for the identification of all isolates to the genus level. The ITS, *tef*, *tub*, *cal* and *his* alignments (including the gaps) were determined to be approximately 570, 470, 450, 610 and 500 bp (base pair) in size, respectively. The combined ITS, *tef*, *tub*, *cal* and *his* sequences of *Diaporthe* contained data for 136 isolates, including the outgroup taxon *Diaporthella corylina* (CBS 121124). The analyses consisted of 31 isolates from this study (Table 1) and 105 sequences (62 type species) originating from GenBank (Table 2). Out of a total of 2594 characters in the MP analyses, 1079 were constant, and 269 were variable and parsimony uninformative. Ten most parsimonious trees resulted from the remaining 1246 parsimony-informative characters (TL = 7439, CI = 0.384, RI = 0.804, RC = 0.309, HI = 0.616). In the ML analyses, the best scoring RAxML tree (Figure 1) with a final likelihood value of −37549.830874 is presented. The matrix had 1675 distinct alignment patterns, with 31.29% of undetermined characters or gaps. Estimated base frequencies were as follows: A = 0.212875, C = 0.328600, G = 0.237362, T = 0.221162; substitution rates AC = 1.075393, AG = 2.704248, AT = 1.155000, CG = 0.851430, CT = 3.774119, GT = 1.000000; gamma distribution shape parameter alpha = 0.464825. The Maximum likelihood (ML) and Bayesian methods (BI) for phylogenetic analyses performed trees with similar topologies.

**Table 2 jof-06-00251-t002:** GenBank accession numbers of species included in the phylogenetic analysis (Figure 1). Ex-type/ex-epitype/ex-isotype/ex-neotype isolates are in bold.

Species Name	Isolate Number	ITS	*tub*	*tef*	*cal*	*his*	Reference
*Diaporthella corylina*	CBS 121124	KC343004	KC343972	KC343730	KC343246	KC343488	Vasilyeva et al. [54]
***Diaporthe acaciarum***	**CBS 138862**	**KP004460**	**KP004509**	**N/A**	**N/A**	**KP004504**	**Crous et al. [55]**
***Diaporthe acuta***	**PSCG 047**	**MK626957**	**MK691225**	**MK654802**	**MK691125**	**MK726161**	**Guo et al. [15]**
*Diaporthe acuta*	PSCG 046	MK626958	MK691224	MK654803	MK691124	MK726162	Guo et al. [15]
***Diaporthe albosinensis***	**CFCC 53066**	**MK432659**	**MK578059**	**MK578133**	**MK442979**	**MK443004**	**Yang et al. [14]**
*Diaporthe albosinensis*	CFCC 53067	MK432660	MK578060	MK578134	MK442980	MK443005	Yang et al. [14]
***Diaporthe ampelina***	**CBS 114016**	**AF230751**	**JX275452**	**AY745056**	**AY230751**	**N/A**	**Mostert et al. [56]**
*Diaporthe ampelina*	CBS 267.80	KC343018	KC343986	KC343744	KC343260	KC343502	Mostert et al. [56]
***Diaporthe angelicae***	**CBS 111592**	**KC343027**	**KC343995**	**KC343753**	**KC343269**	**KC343511**	**Castlebury et al. [57]**
*Diaporthe angelicae*	CBS 100871	KC343025	KC343993	KC343751	KC343267	KC343509	Castlebury et al. [57]
*Diaporthe aquatica*	IFRDCC 3015	JQ797438	N/A	N/A	N/A	N/A	Hu et al. [58]
***Diaporthe aquatica***	**IFRDCC 3051**	**JQ797437**	**N/A**	**N/A**	**N/A**	**N/A**	**Hu et al. [58]**
***Diaporthe araucanorum***	**CBS 145285**	**MN509711**	**MN509722**	**MN509733**	**N/A**	**N/A**	**Zapata et al. [45]**
*Diaporthe araucanorum*	CBS 145284	MN509710	MN509721	MN509732	N/A	N/A	Zapata et al. [45]
***Diaporthe asheicola***	**CBS 136967**	**KJ160562**	**KJ160518**	**KJ160594**	**KJ160542**	**N/A**	**Lombard et al. [59]**
*Diaporthe asheicola*	CBS 136968	KJ160563	KJ160519	KJ160595	KJ160543	N/A	Lombard et al. [59]
*Diaporthe aspalathi*	CBS 117168	KC343035	KC344003	KC343761	KC343277	KC343519	van Rensburg et al. [11]
***Diaporthe aspalathi***	**CBS 117169**	**KC343036**	**KC344004**	**KC343762**	**KC343278**	**KC343520**	**van Rensburg et al. [11]**
***Diaporthe australafricana***	**CBS 111886**	**KC343038**	**KC344006**	**KC343764**	**KC343280**	**KC343522**	**Mostert et al. [56]**
*Diaporthe australafricana*	CBS 113487	KC343039	KC344007	KC343765	KC343281	KC343523	Mostert et al. [56]
*Diaporthe biconispora*	ZJUD61	KJ490596	KJ490417	KJ490475	N/A	KJ490538	Huang et al. [60]
***Diaporthe biconispora***	**ZJUD62**	**KJ490597**	**KJ490418**	**KJ490476**	**KJ490539**	**KJ490539**	**Huang et al. [60]**
***Diaporthe bohemiae***	**CBS 143347**	**MG281015**	**MG281188**	**MG281536**	**MG281710**	**MG281361**	**Guarnaccia et al. [19]**
***Diaporthe caryae***	**CFCC 52563**	**MH121498**	**MH121580**	**MH121540**	**MH121422**	**MH121458**	**Yang et al. [13]**
*Diaporthe caryae*	CFCC 52564	MH121499	MH121581	MH121541	MH121423	MH121459	Yang et al. [13]
***Diaporthe cercidis***	**CFCC 52565**	**MH121500**	**MH121582**	**MH121542**	**MH121424**	**MH121460**	**Yang et al. [13]**
*Diaporthe cercidis*	CFCC 52566	MH121501	MH121583	MH121543	MH121425	MH121461	Yang et al. [13]
***Diaporthe chongqingensis***	**PSCG 435**	**MK626916**	**MK691321**	**MK654866**	**MK691209**	**MK726257**	**Guo et al. [15]**
*Diaporthe chongqingensis*	PSCG 436	MK626917	MK691322	MK654867	MK691208	MK726256	Guo et al. [15]
***Diaporthe cichorii***	**MFLUCC 17-1023**	**KY964220**	**KY964104**	**KY964176**	**KY964133**	**N/A**	**Dissanayake et al. [9]**
***Diaporthe cinnamomi***	**CFCC 52569**	**MH121504**	**MH121586**	**MH121546**	**N/A**	**MH121464**	**Yang et al. [13]**
*Diaporthe cinnamomi*	CFCC 52570	MH121505	MH121587	MH121547	N/A	MH121465	Yang et al. [13]
***Diaporthe cissampeli***	**CPC 27302**	**KX228273**	**KX228384**	**N/A**	**N/A**	**KX228366**	**Crous et al. [61]**
***Diaporthe citri***	**CBS 135422**	**KC843311**	**KC843187**	**KC843071**	**KC843157**	**MF418281**	**Udayanga et al. [6]**
*Diaporthe citri*	AR 4469	KC843321	KC843197	KC843081	KC843167	N/A	Udayanga et al. [6]
***Diaporthe conica***	**CFCC 52571**	**MH121506**	**MH121588**	**MH121548**	**MH121428**	**MH121466**	**Yang et al. [13]**
*Diaporthe conica*	CFCC 52572	MH121507	MH121589	MH121549	MH121429	MH121467	Yang et al. [13]
***Diaporthe coryli***	**CFCC 53083**	**MK432661**	**MK578061**	**MK578135**	**MK442981**	**MK443006**	**Yang et al. [14]**
*Diaporthe coryli*	CFCC 53084	MK432662	MK578062	MK578136	MK442982	MK443007	Yang et al. [14]
***Diaporthe discoidispora***	**ZJUD89**	**KJ490624**	**KJ490445**	**KJ490503**	**N/A**	**KJ490566**	**Huang et al. [60]**
*Diaporthe discoidispora*	ZJUD87	KJ490622	KJ490443	KJ490501	N/A	KJ490564	Huang et al. [60]
***Diaporthe eres***	**AR 5193**	**KJ210529**	**KJ420799**	**KJ210550**	**KJ434999**	**KJ420850**	**Udayanga et al. [7]**
*Diaporthe eres*	CBS 138598	KJ210521	KJ420787	KJ210545	KJ435027	KJ420837	Udayanga et al. [7]
***Diaporthe foikelawen***	**CBS 145289**	**MN509714**	**MN509725**	**MN509736**	**N/A**	**N/A**	**Zapata et al. [45]**
*Diaporthe foikelawen*	CBS 145287	MN509713	MN509724	MN509735	N/A	N/A	Zapata et al. [45]
***Diaporthe fulvicolor***	**PSCG 051**	**MK626859**	**MK691236**	**MK654806**	**MK691132**	**MK726163**	**Guo et al. [15]**
*Diaporthe fulvicolor*	PSCG 057	MK626858	MK691233	MK654810	MK691131	MK726164	Guo et al. [15]
*Diaporthe gulyae*	BRIP 53158	JF431284	KJ197271	N645799	N/A	N/A	Thompson et al. [62]
***Diaporthe gulyae***	**BRIP 54025**	**JF431299**	**KJ197272**	**JN645803**	**N/A**	**N/A**	**Thompson et al. [62]**
***Diaporthe helicis***	**AR 5211**	**KJ210538**	**KJ420828**	**KJ210559**	**KJ435043**	**KJ420875**	**Udayanga et al. [7]**
***Diaporthe hungariae***	**CBS 143353**	**MG281126**	**MG281299**	**MG281647**	**MG281823**	**MG281474**	**Guarnaccia et al. [19]**
*Diaporthe hungariae*	CBS 143354	MG281127	MG281300	MG281648	MG281824	MG281475	Guarnaccia et al. [19]
***Diaporthe juglandicola***	**CFCC 51134**	**KU985101**	**KX024634**	**KX024628**	**KX024616**	**KX024622**	**Yang et al. [18]**
***Diaporthe mahothocarpus***	**CGMCC 3.15181**	**KC153096**	**KF576312**	**KC153087**	**N/A**	**N/A**	**Gao et al. [63]**
*Diaporthe mahothocarpus*	CGMCC 3.15182	KC153097	N/A	KC153088	N/A	N/A	Gao et al. [63]
***Diaporthe malorum***	**CAA734**	**KY435638**	**KY435668**	**KY435627**	**KY435658**	**KY435648**	**Santos et al. [16]**
***Diaporthe millettia***	**GUCC 9167**	**MK398674**	**MK502089**	**MK480609**	**MK502086**	**N/A**	**Long et al. [26]**
***Diaporthe nobilis***	**CBS 587.79**	**KC343153**	**KC344121**	**KC343879**	**KC343395**	**KC343637**	**Li et al. [25]**
***Diaporthe novem***	**CBS 127270**	**KC343155**	**KC344123**	**KC343881**	**KC343397**	**KC343640**	**Santos et al. [64]**
*Diaporthe novem*	CBS 127271	KC343157	KC344125	KC343883	KC343399	KC343641	Santos et al. [64]
***Diaporthe oraccinii***	**LC 3166**	**KP267863**	**KP293443**	**KP267937**	**N/A**	**KP293517**	**Gao et al. [63]**
***Diaporthe osmanthusis***	**GUCC 9165**	**MK398675**	**MK502090**	**MK480610**	**MK502087**	**N/A**	**Long et al. [26]**
***Diaporthe paranensis***	**CBS 133184**	**KC343171**	**KC344139**	**KC343897**	**KC343413**	**KC343655**	**Gomes et al. [4]**
***Diaporthe parvae***	**PSCG 034**	**MK626919**	**MK691248**	**MK654858**	**N/A**	**MK726210**	**Guo et al. [15]**
*Diaporthe parvae*	PSCG 035	MK626920	MK691249	MK654859	MK691169	MK726211	Guo et al. [15]
***Diaporthe pascoei***	**BRIP 54847**	**JX862532**	**KF170924**	**JX862538**	**N/A**	**N/A**	**Tan et al. [65]**
***Diaporthe passiflorae***	**CPC 19183**	**JX069860**	**KY435674**	**KY435633**	**KY435664**	**KY435654**	**Crous et al. [66]**
***Diaporthe patagonica***	**CBS 145291**	**MN509717**	**MN509728**	**MN509739**	**N/A**	**N/A**	**Zapata et al. [45]**
*Diaporthe patagonica*	CBS 145755	MN509718	MN509729	MN509740	N/A	N/A	Zapata et al. [45]
***Diaporthe perjuncta***	**CBS 109745**	**KC343172**	**KC344140**	**KC343898**	**KC343414**	**KC343656**	**van Niekerk et al. [67]**
***Diaporthe phragmitis***	**CBS 138897**	**KP004445**	**KP004507**	**N/A**	**N/A**	**KP004503**	**Crous et al. [55]**
***Diaporthe psoraleae***	**CBS 136412**	**KF777158**	**KF777251**	**KF777245**	**N/A**	**N/A**	**Crous et al. [68]**
***Diaporthe psoraleae-pinnatae***	**CBS 136413**	**KF777159**	**KF777252**	**N/A**	**N/A**	**N/A**	**Crous et al. [68]**
***Diaporthe pterocarpicola***	**MFLUCC 10-0580a**	**JQ619887**	**JX275441**	**JX275403**	**JX197433**	**N/A**	**Udayanga et al. [69]**
*Diaporthe pterocarpicola*	MFLUCC 10-0580b	JQ619887	JX275441	JX275403	JX197433	N/A	Udayanga et al. [69]
***Diaporthe pterocarpi***	**MFLUCC 10-0571**	**JX197433**	**JX275460**	**JX275416**	**JX197451**	**N/A**	**Udayanga et al. [69]**
*Diaporthe pterocarpi*	MFLUCC 10-0575	JQ619901	JX275462	JX275418	JX197453	N/A	Udayanga et al. [69]
***Diaporthe rostrata***	**CFCC 50062**	**KP208847**	**KP208855**	**KP208853**	**KP208849**	**KP208851**	**Fan et al. [17]**
*Diaporthe rostrata*	CFCC 50063	KP208848	KP208856	KP208854	KP208850	KP208852	Fan et al. [17]
***Diaporthe rudis***	**AR 3422**	**KC843331**	**KC843177**	**KC843090**	**KC843146**	**N/A**	**Udayanga et al. [6]**
*Diaporthe rudis*	AR 3654	KC843338	KC843184	KC843097	KC843153	N/A	Udayanga et al. [6]
***Diaporthe sackstonii***	**BRIP 54669b**	**KJ197287**	**KJ197267**	**KJ197249**	**N/A**	**N/A**	**Thompson et al. [70]**
***Diaporthe sennae***	**CFCC 51636**	**KY203724**	**KY228891**	**KY228885**	**KY228875**	**N/A**	**Yang et al. [18]**
*Diaporthe sennae*	CFCC 51637	KY203725	KY228892	KY228886	KY228876	N/A	Yang et al. [18]
***Diaporthe sennicola***	**CFCC 51634**	**KY203722**	**KY228889**	**KY228883**	**KY228873**	**KY228879**	**Yang et al. [18]**
*Diaporthe sennicola*	CFCC 51635	KY203723	KY228890	KY228884	KY228874	KY228880	Yang et al. [18]
***Diaporthe shaanxiensis***	**CFCC 53106**	**MK432654**	**N/A**	**MK578130**	**MK442976**	**MK443001**	**Yang et al. [14]**
*Diaporthe shaanxiensis*	CFCC 53107	MK432655	N/A	MK578131	MK442977	MK443002	Yang et al. [14]
*Diaporthe sojae*	BRIP 54033	JF431295	N/A	KC343901	N/A	N/A	Udayanga et al. [71]
*Diaporthe sojae*	CBS 116019	KC343175	KC344143	KC343901	KC343417	KC343659	Udayanga et al. [71]
*Diaporthe sojae*	FAU 455	KJ590712	KJ610868	KJ590755	KJ612109	KJ659201	Udayanga et al. [71]
***Diaporthe sojae***	**FAU 635**	**KJ590719**	**KJ610875**	**KJ590762**	**KJ612116**	**KJ659208**	**Udayanga et al. [71]**
***Diaporthe spartinicola***	**CPC 24951**	**KR611879**	**KR857695**	**N/A**	**N/A**	**KR857696**	**Crous et al. [71]**
***Diaporthe spinosa***	**PSCG 383**	**MK626849**	**MK691234**	**MK654811**	**MK691129**	**MK726156**	**Guo et al. [15]**
*Diaporthe spinosa*	PSCG 279	MK626925	MK691235	MK654801	MK691126	MK726155	Guo et al. [15]
*Diaporthe subordinaria*	CBS 464.90	KC343214	KC344182	KC343940	KC343456	KC343698	Gomes et al. [4]
*Diaporthe subordinaria*	CBS 101711	KC343213	KC344182	KC343939	KC343455	KC343697	Gomes et al. [4]
***Diaporthe taoicola***	**MFLUCC 16-0117**	**KU557567**	**KU557591**	**KU557635**	**N/A**	**N/A**	**Dissanayake et al. [9]**
***Diaporthe torilicola***	**MFLUCC 17-1051**	**KY964212**	**KY964096**	**KY964168**	**KY964127**	**N/A**	**Dissanayake et al. [9]**
***Diaporthe toxica***	**CBS 534.93**	**KC343220**	**KC344188**	**KC343946**	**KC343462**	**KC343704**	**Williamson et al. [72]**
*Diaporthe toxica*	CBS 546.93	KC343222	KC344190	KC343948	KC343464	KC343706	Williamson et al. [72]
***Diaporthe vangueriae***	**CPC 22703**	**KJ869137**	**KJ869247**	**N/A**	**N/A**	**N/A**	**Crous et al. [55]**
***Diaporthe vawdreyi***	**BRIP 57887a**	**KR936126**	**KR936128**	**KR936129**	**N/A**	**N/A**	**Crous et al. [73]**
***Diaporthe zaobaisu***	**PSCG 031**	**MK626922**	**MK691245**	**MK654855**	**N/A**	**MK726207**	**Guo et al. [15]**
*Diaporthe zaobaisu*	PSCG 032	MK626923	MK691246	MK654856	N/A	MK726208	Guo et al. [15]

AR: Collection of A.Y. Rossman; BRIP: Queensland Plant Pathology herbarium/culture collection, Australia; CBS: Culture collection of the Centraalbureau voor Schimmelcultures, Fungal Biodiversity Centre, Utrecht, The Netherlands; CFCC: China Forestry Culture Collection Center, China; CGMCC: China General Microbiological Culture Collection; CPC: Collection Pedro Crous, housed at CBS; FAU: Isolates in culture collection of Systematic Mycology and Microbiology Laboratory, USDA-ARS, Beltsville, MD, USA; GUCC: Guizhou culture collection, Guizhou, China; IFRDCC: International Fungal Research and Development Centre Culture Collection, Chinese Academy of Forestry, Kunming, China; LC: Corresponding author’s personal collection (deposited in laboratory State Key Laboratory of Mycology, Institute of Microbiology, Chinese Academy of Sciences); MFLUCC: Mae Fah Luang University Culture Collection, Chiang Rai, Thailand; ZJUD: Zhejiang University. ITS, internal transcribed spacers 1 and 2 together with 5.8S nrDNA; *tub*, partial beta-tubulin gene; *cal*, partial calmodulin gene and *tef*, partial translation elongation factor 1-a gene, *his*, histone H3 gene.

The isolates obtained in this study were grouped into twelve clades. Three isolates were grouped with the ex-type of *Diaporthe cercidis* (CFCC 52565) while another three isolates were clustered with the ex-type of *D. nobilis* (CBS 587.79). In addition one isolate with *D. cinnamomi* (CFCC 52569), *D. conica* (CFCC 52571) and *D. sackstonii* (BRIP 54669b) respectively. Twenty-two isolates did not cluster with any known *Diaporthe* species; thus, seven novel species, *Diaporthe constrictospora* (2 isolates, Figure 2), *Diaporthe ellipsospora* (3 isolates, Figure 3), *Diaporthe guttulata* (2 isolates, Figure 4), *Diaporthe irregularis* (4 isolates, Figure 5), *Diaporthe lenispora* (3 isolates, Figure 6), *Diaporthe minima* (4 isolates, Figure 7) and *Diaporthe minusculata* (4 isolates, Figure 8) are determined to be new species based on the morphological and phylogenetic evidence (Figure 1).

### 3.2. Morphology and Culture Characteristics

In this study, thirty-one *Diaporthe* isolates were obtained from decaying woody hosts from six nature reserves in Guizhou province, China (Table 1). The *Diaporthe* isolates obtained in this study were further categorized based on morphological characteristics. Growth was rapid for all isolates grown on PDA, with mycelia covering the whole surface of the Petri dishes. Aerial mycelium was initially white and turned greyish after incubation in the dark at 25 °C for several days. All species exhibited phenotypic characteristics typical of the genus. The seven new species of *Diaporthe* described here are phylogenetically distinct from all previously described species for which sequence data are available.

#### **Taxonomy** 

***Diaporthe constrictospora*** Y.Y. Chen, A.J. Dissanayake and Jian K. Liu ***sp. nov*.** (Figure 2)

*Index Fungorum number*: IF557388; *Facesoffungi Number*: FoF07853; *MycoBank Number*: MB836211

**Etymology**: The epithet from the Latin words constrictus and spora, refers to the slight central constriction often present in the ascospores.

**Holotype**: HKAS 107534

Saprobic on decaying wood. **Sexual morph**: *Ascomata* 190–240 μm diam, black, globose to subglobose or irregular, clustered or solitary, deeply immersed in host tissue. *Asci* 40–48 μm × 9–11 μm ($\bar{x}$ = 43 × 8, n = 30), 8-spored, unitunicate, sessile, elongate to clavate. *Ascospores* 10–12 × 3–4 μm ($\bar{x}$ = 11 × 4, n = 50), hyaline, elongated to elliptical, two-celled, often 4-guttulate, with larger guttules at center and smaller ones at the ends. **Asexual morph**: Not observed.

**Culture characteristics**: Colonies covering entire PDA Petri dishes after 10 d at 25 °C producing abundant white aerial mycelium, reverse fuscous black.

**Material examined: China**, Guizhou Province, Maolan Nature Reserve, saprobic on decaying woody host, April 2017, Y. Y. Chen (HKAS 107534, holotype); ex-type living culture CGMCC 3.20096 = GZCC19-0084; *ibid*, Guiyang District, Huaxi Wetland Park, saprobic on decaying woody host, July 2017, Y. Y. Chen (GZAAS 19-1784, paratype), living culture GZCC 19-0065.

**Notes:** Two strains representing *Diaporthe*
*constrictospora* cluster in a well-supported basal clade (ML/MP/BI = 100/100/1.0) and appear to be distinct from other *Diaporthe* species, and can be easily recognized by their distinctive phylogenetic placement (Figure 1). Since this species is not closely related to any *Diaporthe* species and we were unable to compare the nucleotide differences in the alignment. *Diaporthe constrictospora* is introduced as a phylogenetically distinct species (Figure 1).

***Diaporthe ellipsospora*** Y.Y. Chen, A.J. Dissanayake and Jian K. Liu ***sp. nov*.** (Figure 3)

*Index Fungorum number*: IF557389; *Facesoffungi Number*: FoF07854; *MycoBank Number*: MB836175

**Etymology**: The specific epithet *ellipsospora* refers to the shape of the ascospores.

Holotype: HKAS 107535

Saprobic on decaying branch. **Sexual morph:**
*Ascomata* 380–430 μm diam, black, globose to irregular, scattered on dead twigs, immersed in host tissue, protruding through substrata. *Paraphyses* up to 100 μm long, rarely present, hyaline, smooth, 1–3-septate, cylindrical with obtuse ends, extending above conidiophores. *Asci* 40–47 ×7–8.5 μm ($\bar{x}$ = 43 × 7.5, n = 30), 8-spored, unitunicate, sessile, elongate to clavate. *Ascospores* 8.5–13 × 2.5–3.5 μm ($\bar{x}$ = 10.1 × 3.3, n = 50), hyaline, elongated to elliptical, two-celled, often 4-guttulate, with larger guttules at center and smaller ones at ends. **Asexual morph:** Not observed.

**Culture characteristics**: Colonies on PDA fast growing, covering entire PDA Petri dish after 8 d at 25 °C. White, aerial mycelium turning grey at edges of plate, reverse yellowish pigmentation developing in centre. 

**Material examined: China**, Guizhou Province, Xingyi Wanfenglin, Saprobic on decaying branch, June 2019, Y.Y. Chen (HKAS 107535, holotype), ex-type living culture CGMCC 3.20099 = GZCC 19-0231; *ibid,* (GZAAS 19-2061, paratype), living culture GZCC 19-0342; *ibid*, Maolan Nature Reserve, saprobic on decaying woody host, June 2019, Y. Y. Chen, GZAAS 19-2080, living culture GZCC 19-0357. 

**Notes:***Diaporthe ellipsospora* formed an independent clade (Figure 1) and is phylogenetically distinct from *D. aquatica* in a well-supported clade (ML/MP/BI = 96/100/1.0). *Diaporthe ellipsospora* can be distinguished from *D. aquatica* (IFRDCC 3051) only based on ITS locus (17/539) since other gene sequences (*tef*, *tub*, *cal* and *his*) are unavailable for *D. aquatica*. *Diaporthe ellipsospora* can be morphologically differentiated from *D. aquatica* as the latter has long necks up to 2250 μm [58].

***Diaporthe guttulata*** Y.Y. Chen, A.J. Dissanayake and Jian K. Liu ***sp. nov*.** (Figure 4)

*Index Fungorum number*: IF557390; *Facesoffungi Number*: FoF07855; *MycoBank Number*: MB836212

**Etymology**: Referring to the ascospores with large guttules.

Holotype: HKAS 107536

*Saprobic* on decaying branch. **Sexual morph**: *Ascomata* 560–630 μm diam, black, globose to conical, scattered irregularly, immersed in host tissue with elongated, 300–400 μm long necks protruding through substrata. *Asci* 45–57 μm × 7–9 μm ($\bar{x}$ = 50 × 8, n = 30), unitunicate, 8-spored, sessile, elongate to clavate. *Ascospores* 12–15 × 3–4 μm ($\bar{x}$ = 13 × 3.1, n = 50), elongated to elliptical, hyaline, two-celled, often 4-guttulate, with larger guttules at centre and smaller one at ends. **Asexual morph**: Not observed.

**Culture characteristics**: Colonies covering entire PDA Petri dishes after 7 d at 25 °C producing abundant white aerial mycelium. Reverse white, turning to grey in centre and no conidia produced.

**Material examined: China**, Guizhou Province, Maolan Nature Reserve, saprobic on decaying branch, July 2017, Y.Y. Chen (HKAS 107536, holotype), ex-type living culture CGMCC 3.20100 = GZCC 19-0140; *ibid*, Guiyang District, Suiyang broad water nature reserve, saprobic on decaying woody host, June 2018, Y. Y. Chen (GZAAS 19-2067, paratype), living culture GZCC 19-0371.

**Notes:***Diaporthe guttulata* formed a distinct clade with high support (ML/BI = 88/1.0), and differed with the closely related species *D. angelicae, D. cichorii*, *D. gulyae* and *D. subordinaria*. *Diaporthe guttulata* can be distinguished from *D. angelicae* (7/539 in ITS, 8/467 in *tef*, 7/453 in *tub*, 9/606 in *cal* and 10/513 in *his*); *D. cichorii* (8/539 in ITS, 13/467 in *tef* and 7/453 in *tub* and 21/606 in *cal*); from *D. gulyae* (11/539 in ITS, 8/467 in *tef* and 13/453 in *tub*) and from *D. subordinaria* (6/539 in ITS, 5/467 in *tef*, 15/453 in *tub*, 13/606 in *cal* and 11/513 in *his*). Morphologically, *D. guttulata* differs from *D. cichorii* in having larger asci (50–8 vs. 45–6 μm) and ascospores (13–3 vs. 10–3 μm) [57]. The morphological characters of *Diaporthe guttulata* cannot be compared with *D. gulyae* and *D. subordinaria* as these two species have no reported sexual morphs.

***Diaporthe irregularis*** Y.Y. Chen, A.J. Dissanayake and Jian K. Liu ***sp. nov*.** (Figure 5)

*Index Fungorum number*: IF557391; *Facesoffungi Number*: FoF07856; *MycoBank Number*: MB836213

**Etymology**: Refers to the irregular shape of the ascomata.

Holotype: HKAS 107537

Saprobic on decaying woody branch. **Sexual morph**: *Ascomata* 390–460 μm diam, black, globose to irregular, scattered evenly on dead branches, immersed in host tissue. *Asci* 52–66 × 7–9 μm ($\bar{x}$ = 58 × 8, n = 30), 8-spored, unitunicate, sessile, elongate to clavate. *Ascospores* (10–12 × 3–4 μm ($\bar{x}$ = 11 × 3.5, n = 50), hyaline, two-celled, often 4-guttulate, with larger guttules at center and smaller ones at ends, elongated to elliptical. **Asexual morph**: Not observed.

**Culture characteristics**: Colonies covering entire PDA Petri dishes after 10 d at 25 °C producing abundant white aerial mycelium, reverse fuscous black. 

**Material examined: China**, Guizhou Province, Suiyang broad water nature reserve, saprobic on decaying branch, April 2018. Y.Y. Chen (HKAS 107537, holotype), ex-type living culture CGMCC 3.20092 = GZCC 19-0147; *ibid*., (GZAAS 19-2064, paratype), living culture GZCC 19-0344; *ibid*., GZAAS 19-2069, living culture GZCC 19-0362; *ibid*., GZAAS19-2077, living culture GZCC 19-0352.

**Notes:** Four isolates, representing *Diaporthe irregularis*, are retrieved in a well-supported clade (ML/MP/BI = 100/100/1.0) and appear to be distinct from other *Diaporthe* species phylogenetically (Figure 1). Since this species does not closely related to any particular *Diaporthe* species, we were unable to compare the nucleotide differences in the concatenated alignment. In addition, *Diaporthe irregularis* can be morphologically distinguished from other *Diaporthe* species based on the shape and the position of the ascomata.

***Diaporthe lenispora*** Y.Y. Chen, A.J. Dissanayake and Jian K. Liu ***sp. nov*.** (Figure 6)

*Index Fungorum number*: IF IF557392; *Facesoffungi Number*: FoF07857; *MycoBank Number*: MB836214

**Etymology**: Name reflects the ascospores being smooth-walled, from the Latin lenis referring to smooth and spora.

Holotype: HKAS 107538

Saprobic on decaying woody branch. **Sexual morph**: Ascomata 435–510 μm diam, black, globose to conical, scattered irregularly, immersed in host tissue with elongated, long necks protruding through substrata. *Asci* 44–53 μm × 9–10 μm ($\bar{x}$ = 48 × 9, n = 30), 8-spored, unitunicate, sessile, elongate to clavate. *Ascospores* 10–12 × 2–3 μm ($\bar{x}$ = 11 × 2.5, n = 50), hyaline, two-celled, often 4-guttulate, with larger guttules at centre and smaller one at ends, elongated to elliptical. **Asexual morph**: Not observed.

**Culture characteristics**: Colonies covering entire PDA Petri dishes after 10 d at 25 °C producing abundant white aerial mycelium, reverse early yellow and turned to fuscous black. 

**Material examined: China**, Guizhou Province, Guizhou Province, Suiyang broad water nature reserve, on decaying branch, April 2018. Y.Y. Chen (HKAS 107538, holotype), ex-type living culture CGMCC 3.20101 = GZCC 19-0145; *ibid*., Xingyi Wanfenglin, saprobic on decaying woody host, June 2019, Y. Y. Chen (GZAAS 19-2066, paratype), living culture GZCC 19-0343; ibid., saprobic on decaying branch, June 2018. Y.Y. Chen, GZAAS 19-2075, living culture GZCC19-0351.

**Notes:** In the combined phylogenetic tree, *Diaporthe lenispora* groups in a distinct clade with maximum support (ML/MP/BI = 100/100/1.0) and it appears to be most closely related to *D. vawdreyi* (Figure 1). *Diaporthe lenispora* can be distinguished from *D. vawdreyi* based on ITS, *tef* and *tub* loci (19/539 in ITS, 56/467 in *tef* and 23/453 in *tub*), *cal* and *his* gene regions are unavailable for *D. vawdreyi*. We are not able to compare the morphology of *D. lenispora* and *D. vawdreyi* as the latter has no reported sexual morph [74].

***Diaporthe minima*** Y.Y. Chen, A.J. Dissanayake and Jian K. Liu ***sp. nov*.** (Figure 7)

*Index Fungorum number*: IF557393; *Facesoffungi Number*: FoF07858; *MycoBank Number*: MB836215

**Etymology**: Named for the small conidia.

Holotype: HKAS 107539

*Saprobic* on decaying woody branch. **Sexual morph**: Not observed. **Asexual morph**: *Conidiomata* up to 230 μm in diam., immersed, scattered on PDA, dark brown to black, globose, solitary or clustered in groups of 3–5 conidiomata. *Conidiophores* 9–13 × 1–2 μm ($\bar{x}$ = 11 × 1.5 μm) aseptate, cylindrical, straight or sinuous, densely aggregated, terminal, slightly tapered towards the apex. *Alpha conidia* 6.5–8.5 × 2–3 μm ($\bar{x}$ = 7 × 2 μm), biguttulate, hyaline, fusiform or oval, both ends obtuse. *Beta conidia* not observed.

**Culture characteristics**: Cultures incubated on PDA at 25 °C in darkness. Colony at first flat with white felty mycelium, becoming black in the center and black at the marginal area with 8 d, pycnidia not observed.

**Material examined: China**, Guizhou Province, Guiyang District, Huaxi Wetland Park, Saprobic on decaying branch, April 2017, Y.Y. Chen (HKAS 107539, holotype), ex-type living culture CGMCC 3.20097 = GZCC 19-0066; *ibid*., (GZAAS 19-1786, paratype), living culture GZCC19-0070; *ibid*., GZAAS 19-1787, living culture GZCC19-0061; *ibid*., GZAAS 19-1788, living culture GZCC19-0207.

**Notes:** The phylogenetic result showed that isolates of *Diaporthe minima* clustered closer to *D. bohemiae, D. juglandicola* and *D. rostrata*, and formed a distinct lineage (Figure 1) with maximum support (ML/MP/BI = 100/100/1.0). *Diaporthe minima* can be distinguished from the above closely related species based on ITS, *tef*, *tub*, *cal* and *his* loci for *D. bohemiae* (11/539 in ITS, 45/467 in *tef*, 14/453 in *tub*, 37/606 in *cal*, 34/513 in *his*), *D. juglandicola* (24/539 in ITS, 19/467 in *tef*, 12/453 in *tub*, 27/606 in *cal* and 47/513 in *his*) and *D. rostrata* (19/539 in ITS, 48/467 in *tef*, 13/453 in *tub* 17/606 in *cal* and 49/513 in *his*). Morphologically, *Diaporthe minima* differs from *D. bohemiae, D. juglandicola* and *D. rostrata* in having smaller alpha conidia (7 × 2 vs. 9 × 3 μm) (7 × 2 vs. 11 × 13 μm) [17,18].

***Diaporthe minusculata*** Y.Y. Chen, A.J. Dissanayake and Jian K. Liu ***sp. nov.*** (Figure 8)

*Index Fungorum number*: IF557394; *Facesoffungi Number*: FoF07859; *MycoBank Number*: MB836216

**Etymology**: Name based on a Latin adjective minusculus, meaning rather small conidiomata.

**Holotype**: HKAS 107540

*Saprobic* on decaying branch. **Sexual morph**: Not observed. **Asexual morph**: *Conidiomata* up to 430 μm in diam., superficial, erumpent, scattered on PDA, dark brown to black, globose, solitary or clustered in groups of 3–5 pycnidia, with prominent necks 130–320 μm long. *Conidiophores* 11–18 × 1.5–2.5 μm ($\bar{x}$ = 14 × 2 μm), aseptate, cylindrical, straight or sinuous, densely aggregated, terminal, slightly tapered towards the apex. *Alpha conidia* 7–10 × 2–3 μm ($\bar{x}$ = 9 × 2 μm), biguttulate, hyaline, fusiform or oval, both ends obtuse. Beta conidia not observed.

**Culture characteristics**: Cultures incubated on PDA at 25 °C in darkness showed colony at first white, becoming pale brown with yellowish dots within 10 d, with dense and felted mycelium, visible solitary or aggregated pycnidia at maturity.

**Material examined: China**, Guizhou Province, Xingyi Wanfenglin, Saprobic on decaying branch, June 2019, Y.Y. Chen (HKAS 107540, holotype), ex-type living culture CGMCC 3.20098 = GZCC 19-0215; *ibid*., GZAAS 19-2072, living culture GZCC 19-0372; *ibid*., Suiyang broad water nature reserve, saprobic on decaying woody host, April 2018, Y. Y. Chen (GZAAS19-2062, paratype), living culture GZAAS19-2062; *ibid*., GZAAS19-2070, living culture GZCC 19-0366.

**Notes:** The phylogenetic results showe that *Diaporthe minusculata* clustered close to *D. malorum* and *D. passiflorae*, and formed a distinct lineage (Figure 1) with maximum support (ML/MP/BI = 100/100/1.0). *Diaporthe minusculata* can be distinguished from *D. malorum* (23/539 in ITS, 41/467 in *tef*, 29/453 in *tub*, 52/606 in *cal* and 35/513 in *his*) and from *D. passiflorae* (25/539 in ITS, 13/467 in *tef*, 11/453 in *tub*, 22/606 in *cal* and 17/513 in *his*). Morphologically, *Diaporthe minusculata* differs from *D. malorum* in having conidiomata with a long neck and differs from *D. passiflorae* in shorter conidiophores (14 × 2 vs. 26 × 4 μm) [16,17,18,66,74]. 

## 4. Discussion

Based on the phenotypic characters and the multi-locus phylogeny, the 31 isolates obtained in this study can be recognized as twelve species. Among the five species are previously known and seven species are new to science. These newly discovered species are *Diaporthe constrictospora*, *D. ellipsospora*, *D. guttulata*, *D. irregularis*, *D. lenispora*, *D. minima* and *D. minusculata*. The other taxa are identified as *Diaporthe cercidis* [42], *D. cinnamomi* [42], *D. conica* [42], *D. nobilis* [4] and *D. sackstonii* [70]. Morphological characters of the known species isolated in this study were compared with their original descriptions. Phylogenetically, there were no significant base pair differences between these and their type based combined gene alignments.

A phylogenetic tree derived from an alignment of ITS sequences is beneficial as a guide for the identification of isolates of *Diaporthe* species [65,75]. ITS sequences offer convincing proof for species demarcation where a limited number of taxa are analyzed, such as species associated with the same host [62,64,76]. However, confusion arises when a large number of species from an extensive range of host species are examined. Santos et al. [77] proposed that *tef* is a superior phylogenetic marker in *Diaporthe* than ITS, and has been commonly used as the secondary locus for phylogenetic studies [8,10,64,75]. Gomes et al. [4] studied five loci from 95 species and stated that *tef* poorly distinguished species, and recommended that *his* and *tub* were suitable possibilities as subordinate phylogenetic markers to accompany the authorized fungi barcode: the internal transcribed spacer region (ITS). Dissanayake et al. [10] reviewed the genus *Diaporthe* and provided a checklist for 171 species with available molecular data (from culture and fruiting body) and a phylogenetic tree using four gene regions (ITS, *tef*, *tub* and *cal*). According to Santos et al. [16], incorporation of a five-loci dataset (ITS, *cal*, *his*, *tef*, *tub*) was recommended as the best combination for species identification within the genus and recent studies seems to favor the selection of four or five genes [13,14,15,33,38,39,40,41,42,43,44]. Hence, the present study is conducted combining the five gene regions analyses of ITS, *tef*, *tub*, *cal* and *his* to reveal five known *Diaporthe* species and to assist in the introduction of seven new *Diaporthe* species.

Several studies have been conducted to reveal the association of *Diaporthe* species with various hosts in China. Huang et al. [60] revealed seven apparently undescribed endophytic *Diaporthe* species (*Diaporthe biconispora*, *D. biguttulata*, *D. discoidispora*, *D. multigutullata*, *D. ovalispora*, *D. subclavata* and *D. unshiuensis*) on *Citrus*. Gao et al. [63] identified four novel species (*D. apiculata*, *D. compacta*, *D. oraccinii*, *D. penetriteum*) and three known species (*D. discoidispora*, *D. hongkongensis*, *D. ueckerae*) associated with *Camellia* (tea). Gao et al. [78] showed eight new species of *Diaporthe* (*Diaporthe acutispora*, *D. elaeagni-glabrae*, *D. incompleta*, *D. podocarpi-macrophylli*, *D. undulata*, *D. velutina*, *D. xishuangbanica* and *D. yunnanensis*) from leaves of several hosts while Yang et al. [42] introduced twelve new *Diaporthe* species (*Diaporthe acerigena*, *D. alangii*, *D. betulina*, *D. caryae*, *D. cercidis*, *D. chensiensis*, *D. cinnamomi*, *D. conica*, *D. fraxinicola*, *D. kadsurae*, *D. padina* and *D. ukurunduensis*) from infected forest trees in Beijing, Heilongjiang, Jiangsu, Jiangxi, Shaanxi and Zhejiang Provinces. Three new *Diaporthe* species: *Diaporthe anhuiensis*, *D. huangshanensis*, *D. shennongjiaensis* and two other known species: *D. citrichinensis* and *D. eres* were described as endophytes by Zhou et al. [44]. Yang et al. [14] established three new species: *D. albosinensis*, *D. coryli* and *D. shaanxiensis* isolated from symptomatic twigs and branches at the Huoditang Forest Farm in Shaanxi Province, China. High diversity of *Diaporthe* species associated with pear shoot canker in China was observed by Guo et al. [15] representing thirteen known species (*Diaporthe caryae*, *D. cercidis*, *D. citrichinensis*, *D. eres*, *D. fusicola*, *D. ganjae*, *D. hongkongensis*, *D. padina*, *D. pescicola*, *D. sojae*, *D. taoicola*, *D. unshiuensis* and *D. velutina*) and six new species (*Diaporthe acuta*, *D. chongqingensis*, *D. fulvicolor*, *D. parvae*, *D. spinosa* and *D. zaobaisu*). However, the identification of *Diaporthe* species associated with hosts in nature reserves in China has rarely been studied. Thus, an investigation of *Diaporthe* species was conducted and this provides the first molecular phylogenetic frame of *Diaporthe* diversity in six nature reserves in the Karst region of Guizhou province, combined with morphological descriptions.

Among the twelve species identified in this study, four species have been previously isolated from China. Yang et al. [42] introduced *Diaporthe cercidis* from twigs and branches of *Cercis chinensis* in Jiangsu Province, *D. cinnamomi* from symptomatic twigs of *Cinnamomum* sp. in Zhejiang Province and *D. conica* from symptomatic branches of *Alangium chinense* in Zhejiang Province. *Diaporthe nobilis* has been isolated from *Camellia sinensis* in Guizhou Province [25]. The other known species: *D. sackstonii* [70] has been isolated from petioles of sunflower plants (*Helianthus annuus*) inAustralia. Based on the percentage of occurrence, *Diaporthe irregularis* sp. nov (13%), *D. minima* sp. nov (13%), and *D. minusculata* sp. nov (13%) were categorized as being frequent. *Diaporthe cinnamomi*, *D. conica* and *D. sackstonii* were ranked as infrequent, since only one isolate has been isolated for each species. Interestingly, the type species of the genus, *D. eres* Nitschke [79] was not observed in our survey. This species is one of the frequent species in most of the studies and appears with 365 Fungus–Host combinations [80].

The discovery of these species of *Diaporthe* from different nature reserves in Guizhou province as well as worldwide occurrence shows the polyphagous and cosmopolitan behavior of species in this genus. Certainly, it is obvious that performing complementary studies based on sequencing five gene regions of *Diaporthe* species is essential in order to support reliable species identification. The descriptions and molecular data of *Diaporthe* species provided in this study would serve as a resource for plant pathologists, plant quarantine officials and taxonomists for better identification of *Diaporthe* and its species boundaries. Such studies are necessary to investigate this group of fungi in different unexploited biomes, to reveal the degree of diversity and to support more suitable control measures to prevent their dissemination. Importantly, based on the *Diaporthe* taxa identification in this study coupled with previous studies, it could be concluded that almost all the known species isolated (*Diaporthe cercidis*, *D. cinnamomi*, *D. conica*, *D. nobilis* and *D. sackstonii*) as saprobes in this study were pathogenic on various host plants [25,42,70]. This could indicate that the seven newly introducing species could potentially be pathogens even though they were isolated from decaying woody hosts, and their pathogenicity should be evaluated in further studies with more samples (from other kinds of habitats and hosts, as well as the different distributions and substrates). In the meantime, we provided the culture details and deposited them in publicly accessible culture collections for further evaluation or comparison of the life modes of these taxa.

## 5. Conclusions

We carried out fungal diversity investigations with large-scale sampling in the Karst region of southwestern China and this is the first report of *Diaporthe* species isolated from nature reserves in Karst region of Guizhou province, China. The identification of twelve *Diaporthe* species (five known species and seven new species) associated with saprobic woody hosts is documented.

## Figures and Tables

**Figure 1 jof-06-00251-f001:**
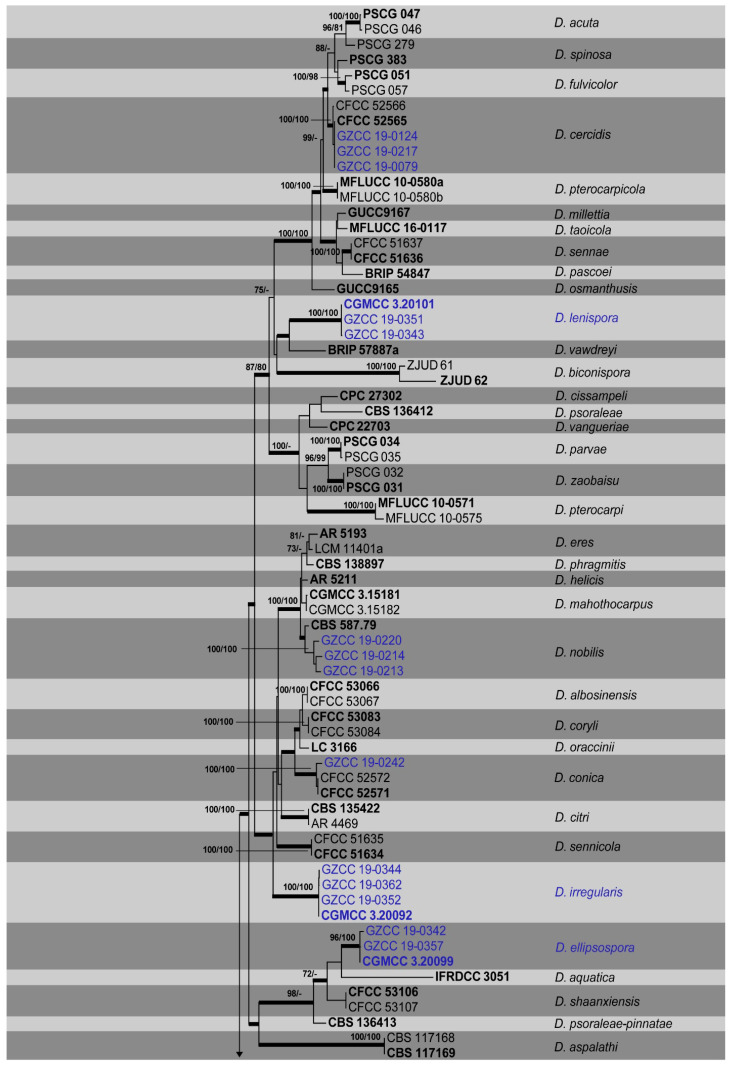
Phylogram generated from maximum likelihood analysis of *Diaporthe* species isolated in this study and their phylogenetically closely related species based on combined internal transcribed spacer region (ITS), *tef*, *tub*, *cal* and *his* sequence data. Bootstrap support values for ML ≥ 70%, MP ≥ 70%, are indicated above the nodes and the branches are in bold indicate Bayesian posterior probabilities ≥0.9. The tree is rooted with *Diaporthella corylina* (CBS 121124). Isolate numbers of ex-types and reference strains are in bold. Taxa isolated in this study are in blue.

**Figure 2 jof-06-00251-f002:**
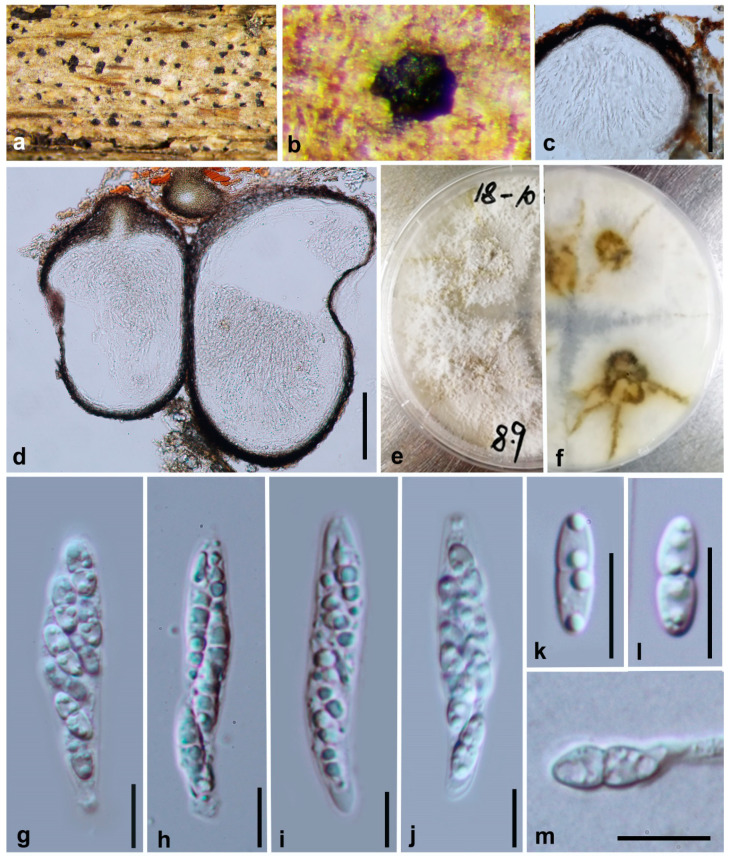
***Diaporthe constrictospora*** (HKAS 107534, holotype). (**a**,**b**) Ascomata on host surface. (**c**,**d**) Section ofascomata (**e**) 10 days old culture on potato dextrose agar (-) from above. (**f**) 10 days old culture on PDA from reverse. (**g**–**j**) Asci. (**k**,**l**) Ascospores. (**m**) Germinating ascospore. Scale bars: (**c**,**d**) = 100 μm, (**g**–**m**) = 10 μm.

**Figure 3 jof-06-00251-f003:**
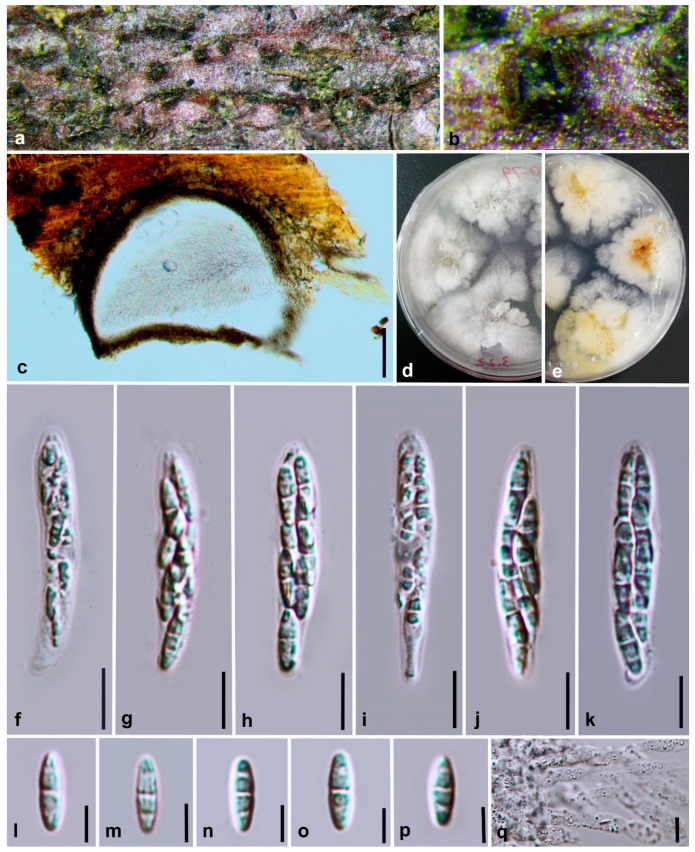
***Diaporthe ellipsospora*** (HKAS 107535, holotype). (**a**,**b**) Ascomata on host surface. (**c**) Section of an ascoma. (**d**) 8 days old culture on PDA from above. (**e**) 8 days old culture on PDA from reverse. (**f**) Immature ascus. (**g**–**k**) Mature asci. (**l**–**p**) Ascospores. (**q**) Paraphyses. Scale bars: (**c**) = 100 μm, (**f**–**k**) = 10 μm, (**l**–**q**) *=* 5 μm.

**Figure 4 jof-06-00251-f004:**
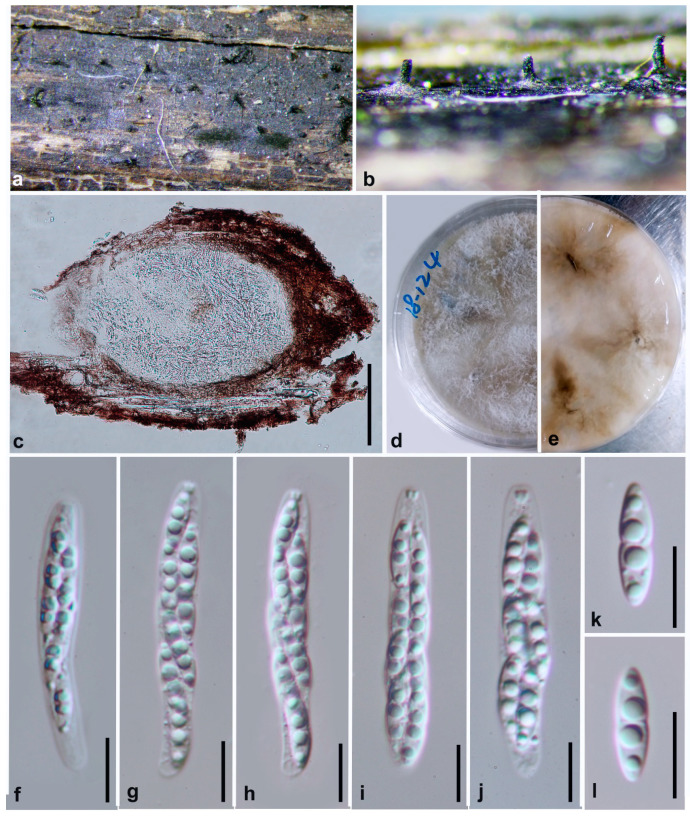
***Diaporthe guttulata*** (HKAS 107536, holotype). (**a**,**b**) Ascomata on host surface. (**c**) Section of an ascoma. (**d**) 7 days old culture on PDA from above. (**e**) 7 days old culture on PDA from reverse. (**f**) Immature ascus. (**g**–**j**) Mature asci. (**k**,**l**) Ascospores. Scale bars: (**c**) = 100 μm, (**f**–**l**) = 10 μm.

**Figure 5 jof-06-00251-f005:**
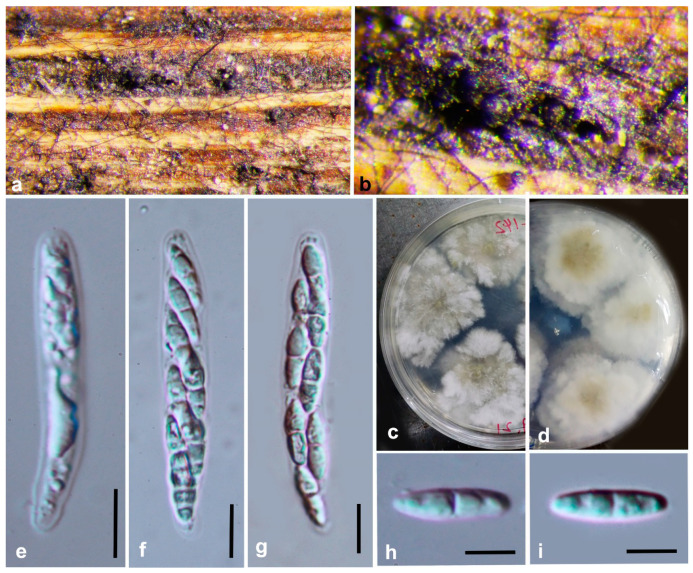
***Diaporthe irregularis*** (HKAS 107537, holotype). (**a**,**b**) Ascomata on host surface. (**c**) 10 days old culture on PDA from above. (**d**) 10 days old culture on PDA from reverse. (**e**) Immature ascus. (**f**,**g**) Mature asci. (**h**,**i**) Ascospores. Scale bars: (**e**–**g**) = 10 μm, (**h**,**i**) = 5 μm.

**Figure 6 jof-06-00251-f006:**
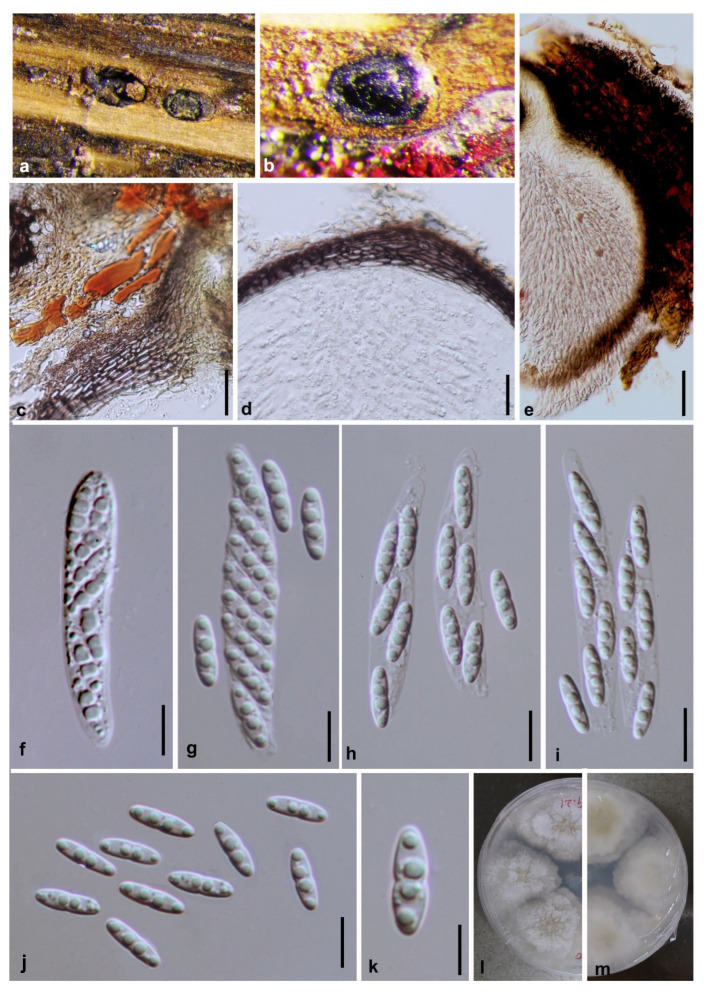
***Diaporthe lenispora*** (HKAS 107538, holotype). (**a**,**b**) Ascomata on host surface. (**c**) Ostiole. (**d**,**e**) Section of ascomata. (**f**) Immature ascus. (**g**–**i**) Mature asci. (**j**,**k**) Ascospores. (**l**) 10 days old culture on PDA from above. (**m**) 10 days old culture on PDA from reverse. Scale bars: (**c**) = 100 μm, (**d**,**e**) = 50 μm, (**f**–**j**) = 10 μm, (**k**) = 5 μm.

**Figure 7 jof-06-00251-f007:**
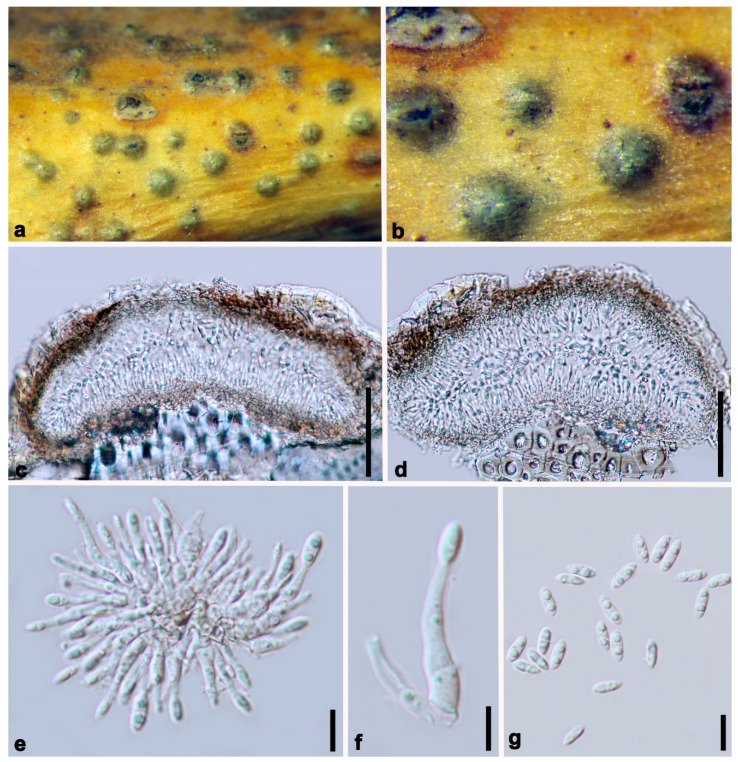
***Diaporthe minima*** (HKAS 107539, holotype). (**a**,**b**) Conidiomata on host surface. (**c**,**d**) Section of conidiomata. (**e**,**f**) Alpha conidia attached to conidiogenous cells. (**g**) Alpha conidia. Scale bars: (**c**,**d**) = 50 μm, (**e**–**g**) = 10 μm.

**Figure 8 jof-06-00251-f008:**
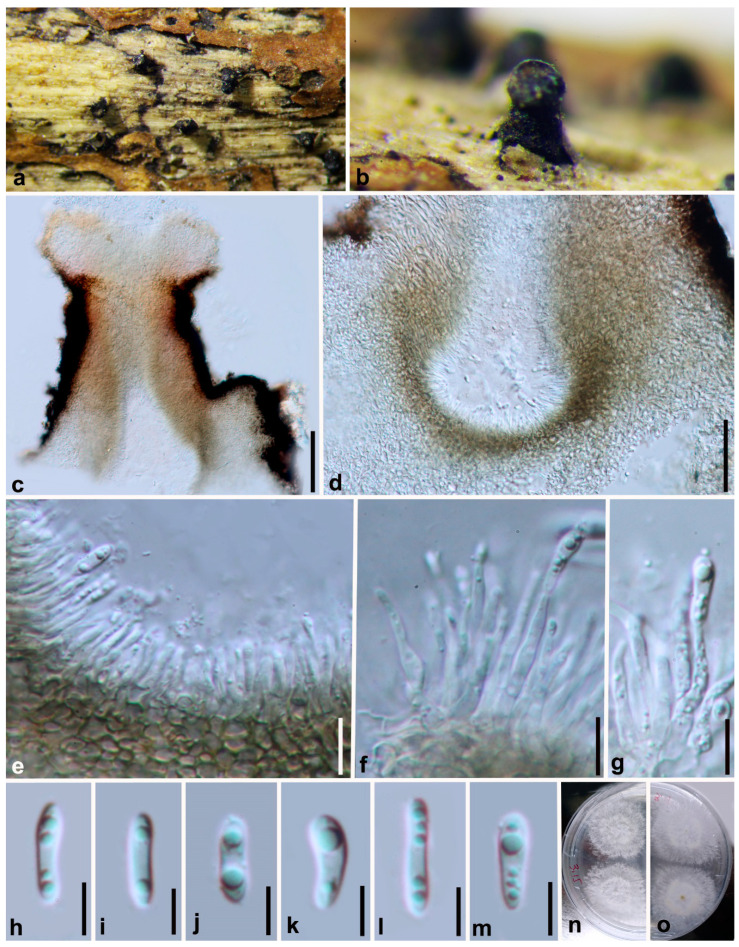
***Diaporthe minusculata*** (HKAS 107540, holotype). (**a**,**b**) Conidiomata on host surface. (**c**–**e**) Section of conidiomata. (**f**,**g**) Alpha conidia attached to conidiogenous cells. (**h**–**m**) Alpha conidia. (**n**) 5 days old culture on PDA from above. (**o**) 5 days old culture on PDA from reverse. Scale bars: (**c**) = 100 μm, (**d**) = 50 μm, (**e**–**g**) = 10 μm, (**h**–**m**) = 5 μm.

**Table 1 jof-06-00251-t001:** *Diaporthe* species studied in this study (Figure 1). Details of ex-type species introduced in this study are in bold.

Species	Isolate	Locality	ITS	*tef*	*tub*	*cal*	*his*
*Diaporthe cercidis*	GZCC 19-0079	Guiyang Xiaochehe Wetland Park	MT385942	MT424677	MT424698	MW022466	MW022482
*D. cercidis*	GZCC 19-0124	Maolan Nature Reserve	MT385943	MT424678	MT424699	MW022467	MW022483
*D. cercidis*	GZCC 19-0217	Xingyi Wanfenglin	MT385944	MT424679	MT424700	MW022468	MW022484
*D. cinnamomi*	GZCC 19-0274	Maolan Nature Reserve	MT385945	MT424680	N/A	MT424717	MW022485
*D. conica*	GZCC 19-0242	Maolan NatureReserve	MT385946	MT424681	MT424701	MW022469	MW022486
***D. constrictospora***	**CGMCC 3.20096**	**Maolan Nature Reserve**	**MT385947**	**MT424682**	**MT424702**	**MT424718**	**MW022487**
*D. constrictospora*	GZCC 19-0065	Guiyang Huaxi Wetland Park	MT385948	MT424683	MT424703	MT424719	N/A
***D. ellipsospora***	**CGMCC 3.20099**	**Xingyi Wanfenglin**	**MT385949**	**MT424684**	**MT424704**	**MT424720**	**MW022488**
*D. ellipsospora*	GZCC 19-0342	Xingyi Wanfenglin	MT797176	MT793019	MT793030	MT786247	MW022489
*D. ellipsospora*	GZCC 19-0357	Maolan Nature Reserve	MT797177	MT793020	MT793031	MT786248	MW022490
***D. guttulata***	**CGMCC 3.20100**	**Maolan Nature Reserve**	**MT385950**	**MT424685**	**MT424705**	**MW022470**	**MW022491**
*D. guttulata*	GZCC 19-0371	Suiyang water nature reserve	MT797178	MT793021	MT793032	MW022471	MW022492
***D. irregularis***	**CGMCC 3.20092**	**Suiyang water nature reserve**	**MT385951**	**MT424686**	**MT424706**	**MT424721**	**N/A**
*D. irregularis*	GZCC 19-0344	Suiyang water nature reserve	MT797179	MT793022	MT793033	MT786249	N/A
*D. irregularis*	GZCC 19-0362	Suiyang water nature reserve	MT797180	MT793023	MT793034	MT786250	N/A
*D. irregularis*	GZCC 19-0352	Suiyang water nature reserve	MT797181	MT793024	MT793035	MT786251	N/A
***D. lenispora***	**CGMCC 3.20101**	**Suiyang water nature reserve**	**MT385952**	**MT424687**	**MT424707**	**MW022472**	**MW022493**
*D. lenispora*	GZCC 19-0343	Xingyi Wanfenglin	MT797182	MT793025	MT793036	MW022473	MW022494
*D. lenispora*	GZCC 19-0351	Xingyi Wanfenglin	MT797183	MT793026	MT793037	MW022474	MW022495
***D. minima***	**CGMCC 3.20097**	**Guiyang Huaxi Wetland Park**	**MT385953**	**MT424688**	**MT424708**	**MT424722**	**MW022496**
*D. minima*	GZCC 19-0070	Guiyang Huaxi Wetland Park	MT385954	MT424689	MT424709	MT424723	MW022497
*D. minima*	GZCC 19-0061	Guiyang Huaxi Wetland Park	MT385955	MT424690	MT424710	MT424724	MW022498
*D. minima*	GZCC 19-0207	Guiyang Huaxi Wetland Park	MT385956	MT424691	MT424711	MT424725	N/A
***D. minusculata***	**CGMCC 3.20098**	**Xingyi Wanfenglin**	**MT385957**	**MT424692**	**MT424712**	**MW022475**	**MW022499**
*D. minusculata*	GZCC 19-0345	Suiyang water nature reserve	MT797184	MT793027	MT793038	MW022476	MW022500
*D. minusculata*	GZCC 19-0366	Suiyang water nature reserve	MT797185	MT793028	MT793039	MW022477	MW022501
*D. minusculata*	GZCC 19-0372	Xingyi Wanfenglin	MT797186	MT793029	MT793040	MW022478	MW022502
*D. nobilis*	GZCC 19-0213	Fanjing mountain	MT385958	MT424693	MT424713	MT424726	MW022503
*D. nobilis*	GZCC 19-0220	Xingyi Wanfenglin	MT385959	MT424694	MT424714	MW022479	MW022504
*D. nobilis*	GZCC 19-0214	Fanjing mountain	MT385960	MT424695	MT424715	MW022480	MW022505
*D. sackstonii*	GZCC 19-0129	Maolan Nature Reserve	MT385962	MT424697	MT424716	MT424727	MW022507

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
