# Peer review of "Unravelling Diaporthe Species Associated with Woody Hosts from Karst Formations (Guizhou) in China"

_jof, 2020, doi:10.3390/jof6040251_

Round 1

Reviewer 1 Report

-According to the Journal instructions, abbreviations have to be explained not only in the abstract, but also in the introduction and in Tables and Figures when they are mentioned for the first time. For instance, ITS and others.

-Lines 41-51: This is a confusing sentence. Please, rewrite it.

-Lines 99-100: Please, indicate the concentrations of the PCR reagents and not only the volumes.

-PAUP description appears in lines 122-123, but PAUP is mentioned for the first time in line 113. Please, put this description above.

-line 123: MP analysis is repeated here. It is already described before, line 114.

-Line 130: BMCMC should be MCMC.

-The phylogenetic tree is incomplete. Please, introduce a complete one in which the new species and the outgroup are visible. It is not possible to evaluate the results without this information. Moreover, could you please indicate the meaning of the band separation with different grey colors in this phylogenetic tree?

-In the results section there are no comments about “ambiguous regions excluded” neither KHT wich are mentioned in lines 123 and 128 of the materials and methods section, respectively. Please, add these comments.

-Line 148: The length of the TEF sequence with these primers is probably shorter. This size may correspond to that of the alignment not the sequence.

-Line 150: The total number of isolates that appear in the tables is 132 (100 references, including the outgroup, and 32 from this study), not 136. Please, revise it.

-Line 153-155: The presented tree is from ML analyses not the parsimony tree

-Line 172: It should be “Bootstrap support values for ML≥70 %, MP≥70 %”, not Parsimony bootstrap

-Line 289: (BI/ML/MP = 98/99/1.0) this order is not correct.

-Line 315: (ML/MP/BPP=96/99/1). Sometimes you use BI, instead of BPP. Please uniformize it in all the text.

-Line 369: having smaller, not havingsmaller

Author Response

Dear Reviewer,

Thank you for the review and suggestions made for improving this manuscript. We really appreciate it and have considered all of the major changes. A revised manuscript has been prepared. Responses to reviewers’ comments (point to point) are indicated as required.

Response to Reviewer1:

Point 1: According to the Journal instructions, abbreviations have to be explained not only in the abstract, but also in the introduction and in Tables and Figures when they are mentioned for the first time. For instance, ITS and others.

Response 1: We agree with your comment. The abbreviations were explained when it is mentioned for the first time. We checked the entire manuscript including the tables and figures, and corrected where necessary.

Point 2: Lines 41-51: This is a confusing sentence. Please, rewrite it.

Response 2: We have re-written the sentences starting from Line 41 and Line 51.

Point 3: Lines 99-100: Please, indicate the concentrations of the PCR reagents and not only the volumes.

Response 3: Thank you very much for your comment. We included the concentrations of reagents in the lines 99-101.

Point 4: PAUP description appears in lines 122-123, but PAUP is mentioned for the first time in line 113. Please, put this description above.

Response 4: Thank you so much. We corrected this mistake in the manuscript. The PAUP abbreviation was explained in the line 116.

Point 5: line 123: MP analysis is repeated here. It is already described before, line 114.

Response 5: Yes, thank you for your comment. We avoided this repetition in the lines 123-124.

Point 6: Line 130: BMCMC should be MCMC.

Response 6: We changed the BMCMC in to MCMC in line 130.

Point 7: The phylogenetic tree is incomplete. Please, introduce a complete one in which the new species and the outgroup are visible. It is not possible to evaluate the results without this information. Moreover, could you please indicate the meaning of the band separation with different grey colors in this phylogenetic tree?

Response 7: We are sorry that the second part of the Figure 1 was missing in the review version. We have checked again the submit version and it was a completed Figure (two pages), the second part was missed when the journal prepared the review version. We have provided the updated (5 gene regions followed the reviewer suggestions) phylogenetic tree (Figure 1) in the revised version.

Thanks very much for your suggestion and we have improved the Figure 1 and the band separation has been done for each species of the genus.

Point 8: In the results section there are no comments about “ambiguous regions excluded” neither KHT which are mentioned in lines 123 and 128 of the materials and methods section, respectively. Please, add these comments.

Response 8: Thank you for the comment. As we have included this in the Materials and methods section we did not include it to the results section.

Point 9: Line 148: The length of the TEF sequence with these primers is probably shorter. This size may correspond to that of the alignment not the sequence.

Response 9: We agree with your comment. The length of the TEF gene region (TEF728F/TEF986R) is approximately 300bp. When aligning with more sequences, more gaps are automatically originated. That’s the reason for 440 bp length in TEF. Hence we changed the word ‘sequence’ to the word ‘alignments (including the gaps)’ in the line 149.

Point 10: Line 150: The total number of isolates that appear in the tables is 132 (100 references, including the outgroup, and 32 from this study), not 136. Please, revise it.

Response 10: We have revised this and the total amount of isolates included in both phylogenetic analysis and the revised table are 136. We have included the missing isolate details to the Table 2.

Point 11: Line 153-155: The presented tree is from ML analyses not the parsimony tree

Response 11: Thank you for this comment. We corrected this in the manuscript by changing the lines 150-152. The presented tree is from ML analyses.

Point 12: Line 172: It should be “Bootstrap support values for ML≥70 %, MP≥70 %”, not Parsimony bootstrap

Response 12: Thank you for the comment. We removed the word ‘Parsimony’ and included it as ‘Bootstrap support values for ML≥70 %, MP≥70 %’ in the line 185.

Point 13: Line 289: (BI/ML/MP = 98/99/1.0) this order is not correct.

Response 13: The order was corrected in the manuscript as (ML/MP/BI= 98/99/1.0) in the line 334.

Point 14: Line 315: (ML/MP/BPP=96/99/1). Sometimes you use BI, instead of BPP. Please uniformize it in all the text.

Response 14: Sorry for this mistake. The line 360 was corrected as (ML/MP/BI= 96/99/1.0). The entire manuscript was checked and ‘BPP’ was replaced with ‘BI’.

Point 15: Line 369: having smaller, not havingsmaller

Response 15: Thank you for pointing out this. We corrected it as ‘having smaller’ in the line 414.

Best regards,

Jian-Kui Liu

Reviewer 2 Report

The manuscript by Dissanayake et al., aims to describe and illustrate the Diaporthe taxa from natural reserves in Karst region of Guizhou province based on morphological characters and phylogenies derived from combined ITS, TEF, TUB and CAL gene sequences. The provide some evidence of the existence of 7 new species of Diaporthe. In general, this paper presents some issues that deserve attention.

  1. Despite, in the introduction, the authors recognized that for the Diaporthe phylogenetic analysis the most advisable is to sequence 5 genes (ITS, TEF, TUB, histone, CAL), they only sequenced 4 (ITS, TEF, TUB, and CAL ). Why?
  2. After reading the entire article I still do not know: What is the importance of work?Why did the survey take place in the natural reserve?What is this work really for, just to find new species?!!
  3. The authors affirm that they describe 7 new species based on the morphological characteristics and on the sequences of the 4 genes (ITS, TEF, TUB, and CAL ), but for 3 species they only use ITS, TEF and TUB.
  4. In my opinion, by using 4 genes trees as backbone trees for the phylogenetic analysis the authors will be under looking the Diaporthe species that may be related to the new species.To improve this analysis, an ITS or TEF tree should be made, with all available sequences of the Diaporthe type species.

The authors should pay attention when submitting an article, be careful to send the complete figures (trees), submit good quality photos (photos of the species description). And furthermore, be precise, professionals and always make a revision in the databases that you present, so there is not missing sequence essential for your analysis (Table 2). In the attached pdf you can see more comments and suggestions.

Author Response

Dear Reviewer,

Thank you for the review and suggestions made for improving this manuscript. We really appreciate it and have considered all of the major changes. A revised manuscript has been prepared. Responses to reviewers’ comments (point to point) are indicated as required.

Response to Reviewer 2:

Point 1: The manuscript by Dissanayake et al., aims to describe and illustrate the Diaporthe taxa from natural reserves in Karst region of Guizhou province based on morphological characters and phylogenies derived from combined ITS, TEF, TUB and CAL gene sequences. They provide some evidence of the existence of 7 new species of Diaporthe. In general, this paper presents some issues that deserve attention.

Response 1: Thank you very much for your insightful comments as they really helped us to improve the manuscript.

Point 2: Despite, in the introduction, the authors recognized that for the Diaporthe phylogenetic analysis the most advisable is to sequence 5 genes (ITS, TEF, TUB, histone, CAL), they only sequenced 4 (ITS, TEF, TUB, and CAL ). Why?

Response 2: We agree with your comment and we sequenced the HIS gene region for the isolates obtained in this study (except for D. irregularis). We attempted to generate the sequence of HIS gene region for the four isolates of D. irregularis, but it was not success. However we got the sequences of the HIS gene region for other 28 isolates and included them in the phylogenetic analyses.

Point 3: After reading the entire article I still do not know: What is the importance of work? Why did the survey take place in the natural reserve? What is this work really for, just to find new species?!!

Response 3: Thank you for concerns regarding the contribution of our study.

Karst formations in China perform as a unique landform, comprising a rich biodiversity of flora and fauna. Earlier, there was a shortage of investigations of fungi related researches in this distinctive landform area. We are carrying out fungal diversity investigations with large scale in the Karst region of southwestern China. In this study we are dealing with fungi in Karst regions located in Guizhou province, China, especially in in several natural reserves (the brief information is provided in the second paragraph of Introduction). . In recent years, our work has been carried out as a series of findings/publications. In these series, we have already identified various kinds of fungi from this Karst region and published them in relevant journals. However, during these investigations, more Diaporthe isolates have been collected from the natural reserves which made our great attention. Hence, we treated those isolates incorporating both morphology and molecular phylogeny and realized that many species are yet to be discovered from the Karst formations. We assume this study would serve as the base for more fungi-related studies in this Karst region in future.

We also address your concern in the last paragraph of discussion by adding the following sentences “Importantly, based on the Diaporthe taxa identification in this study coupled with previous studies, it could be concluded that almost all the known species isolated (Diaporthe cercidis, D. cinnamomi, D. conica, D. nobilis, D. pascoei and D. sackstonii) as saprobes in this study were pathogenic on various host plants. This could indicate that the seven newly introducing species can be potentially pathogens even though they were isolated from decaying woody hosts, and their pathogenicity should be evaluated in further studies with more sample investigations (from other kinds of habitats and hosts, as well as the different distributions and substrates). In the mean time we have provided the culture details and deposited them in publicly accessible culture collections for further evaluation or comparison of the life modes of these taxa.”

Point 4: The authors affirm that they describe 7 new species based on the morphological characteristics and on the sequences of the 4 genes (ITS, TEF, TUB, and CAL), but for 3 species they only use ITS, TEF and TUB.

Response 4: Thank you for this comment. We would like to reply that we were able to sequence the HIS gene region for 28 isolates except the four isolates which belong to D. irregularis (we have tried several times and still not able to generate the results for this species)

Also we could sequence the CAL gene region for those three new species which lack the CAL sequences earlier.

These newly generated sequence data were included to the phylogenetic analyses.

Point 5: In my opinion, by using 4 genes trees as backbone trees for the phylogenetic analysis the authors will be under looking the Diaporthe species that may be related to the new species. To improve this analysis, an ITS or TEF tree should be made, with all available sequences of the Diaporthe type species.

Response 5: Thank you for this comment. Yes we constructed an ITS tree with all the type species (if the sequences are available). This tree is provided as a supplementary material.

Point 6: The authors should pay attention when submitting an article, be careful to send the complete figures (trees), submit good quality photos (photos of the species description). And furthermore, be precise, and always make a revision in the databases that you present, so there is not missing sequence essential for your analysis (Table 2). In the attached pdf you can see more comments and suggestions.

Response 6: We apologize for those mistakes we have in the submission version. With your comments, we have improved the entries, plates and completed tables in the manuscript. Thank you so much for your valuable comments and suggestions. Especially herewith we acknowledge your humble words which really helped to improve the manuscript.

Reponses to the comments on the manuscript by Reviewer 2 (with the PDF tracking)

Point 1: Line 25

Revised the word ‘nature’as ‘natural’ in the line 26.

Response 1: We have checked it and it can be either “nature” or “natural” which the “nature” is used more frequent. Hereby we keep use “nature” instead of changing to “natural”

Point 2: Line 60 – why not 5 genes?

Response 2: HIS gene region was sequenced for all isolates except the four isolates of D. irregularis.

CAL gene region was sequenced for the other three new species.

Hence, five gene regions: ITS, TEF, TUB, CAL and HIS were incorporated in the phylogenetic analyses. This was mentioned in the line 60.

Point 3: Line 79 – Should be at least 100 conidia

Response 3: Thanks very much for raising this and we have calculated it again for all of our species with 100 and the average measurements of the conidia have no much changes and the descriptions provided the amount in papers are 30-50 in general, and we have revised them as 50.

Point 4: Line 91 - How did you made this identification? inhouse database, genbank, fingerprinting technique ,...?

Response 4: For the identification of Diaporthe, ITS was sequenced for all isolates and BLAST search (basic local alignment search tool) at GenBank was used to reveal the closest matching taxa. These details were included to the line 91.

Point 5: Line 108 - In this way it is not possible to see all the species that relate to your species. This tree should be given as supplementary material

Response 5: As of the answer given to the above 4th question, we will provide an ITS tree (with all type/ex-type/neo-type Diaporthe species from previous studies) as a supplementary material, please kindly check it with the revised version.

Point 6: Line 138 - only the MP tree is available...

Response 6: We have submitted a new ML tree to the treebase. The accession number has been provided in the manuscript together with the reviewer URL in the line 138.

Point 7: Line 150 – How many type species? 136 isolates are from this study?

Response 7: We have revised the line 150 in the manuscript. 104 isolates were obtained from the GenBank (including the outgroup). Among these, 63 isolates were type species. 32 isolates were obtained from this study.

Point 8: Line 156 - (Bayesian tree not shown). Why?

Response 8: Thank you for this comment. Since the topologies of both ML tree and BI tree are similar, we provided only the ML tree as presented backbone phylogenetic tree in the manuscript and add the values of other two methods’ results.

Point 9: Line 163 – Table 1 incomplete

Response 9: We have sequenced the CAL and HIS gene regions, so the Table 1 can be filled with the new GenBank accession numbers. We have submitted the sequences to the GenBank and waiting for the accession numbers.

Point 10: Line 164 - new species described based only in 3 genes: ITS, TEF and TUB!!!!!! Not acceptable

Response 10: We sequenced the HIS gene region and CAL gene region for other three new species. So this problem was solved.

Point 11: Line 167 – TUB available

Response 11: This was corrected in the Table 2 by adding the TUB sequences of D. gulyae and D. spartinicola.

Point 12: Line 168 - only MP and ML values are marked in the figure...and the Bayesian values?

Response 12: We have made thickened branches for The Bayesian values were indicted by thickened branches in the phylogenetic tree.

Point 13: Line 169 - the figure is incomplete.... this is only half the tree.

Response 13: The completed figure 1 was included to the manuscript.

Point 14: Line 189 - is this correct? scale bars from o, p and q are 10 um?

Response 14: Yes. The ascospore size varies from 10-12 um.

Point 15: Bad quality photos

Response 15: The plate has been improved.

Point 16: Line 199 - caption incorrect

Response 16: All figure captions were corrected in the manuscript.

Point 17: Line 219 - the photos from the plates were taken when? 5 days of growth. 10 days? You should say it, in the caption of the figures.

Response 17: The figure captions were changed according to the number of days for the PDA plates. Normally they varied from 5 to 10 days.

Point 18: Line 235 - Are these observations made at the same time of the photos of the figures?

Response 18: Yes. It’s a 10 day old culture.

Point 19: Line 241 - tree incomplete, for that is impossible to compare

Response 19: The updated Figure has been provided.

Point 20: Line 243 – TUB sequence is available.

Response 20: Thank you for pointing out this. We have included the TUB sequences of D. spartinicola to the Table 1. The TUB sequence differences of both D. constrictospora and D. spartinicola were compared. The line 225 was revised accordingly.

Point 21: Line 291 - Are this differences enough to differentiate species? It could be intraspecific variation. And the differences in CAL sequences?

Response 21: Since Diaporthe species are difficult to differentiate using morphological characters we have to mostly rely on molecular sequence data and the topology of the phylogenetic analysis.  Hence the base pair differences and distinct topologies of the clades would serve as the species limits.

CAL base pair differences were included to the manuscript (line 336).

Point 22: Regarding to D. gulyae is missing the comparison of TUB, and for D. suborninaria with CAL.

Response 22: We have included and compared the TUB base pair variations of D. gulyae with D. guttulata (Line 338). Also we are able to compare the CAL and HIS base pair differences of D. subordinaria and D. guttulata (line 339).

Point 23: Line 392 - statement not true, D. passiflorae have all 5 genes (ITS, TEF, TUB, HIS, CAL) sequenced

Response 23: We have included all five sequence data (ITS, TEF, TUB, CAL and HIS) for D. passifloare in the Table 2. The base pair comparisons have been made in the manuscript, line 439.

Point 24: Line 405 - If so, why didn´t you did one with all ITS available sequences?

Response 24: A phylogenetic tree derived from an alignment of ITS sequences is provided as a supplementary material.

Point 25: Line 454 - Why? in my opinion it would be important for plant pathologists and plant quarantine officials if we made some pathogenicity tests.

Response 25: Yes. Your comment is some far correct. Since the isolates obtained in this study are saprobic, the findings would not directly relate to the plant pathologists or plant quarantine officials. However, in future if these species would found as pathogenic strains in commercially important crops/hosts, then there will be an interest for plant pathologists to find out their life modes. Hence, they can check and compare both saprobic and pathogenic modes of particular species. This is also addressed in the discussion part.

Best regards,

Jian-Kui Liu

Round 2

Reviewer 2 Report

Please do a thorough review of table 2. When building this type of table, you must be very thorough and accurate since these data are as important or more important than obtaining good quality sequences of the studied isolates. The lack of inconsistency in the data in this table demonstrates the lack of professionalism, lack of consideration by the reviewer and the other researchers who may have access to this article.

Figure 1 must be built again, with the correct data. I suggest redoing the trees of 5 genes, using species which I pointed out in gray in PDF supplementary material.

Regarding the nucleotide’s variations, D. guttalata is really D. angelicae, and not a new specie.

In the attached pdf you can see more comments and suggestions.

Author Response

Dear Reviewer,

Thanks very much for your effort to improve our manuscript and we have learned a lot by revising the paper following your comments. 

The multi-gene phylogenetic tree is revised by adding more taxa (with your suggestions), the Table 2 has been checked again and any necessary correction have been made (we are sorry for the mistakes we had and you are absolutely right, it is quite important to check carefully and provide the correct information for these taxa and isolates). 

We really appreciate your comments and suggestion to improve our manuscript. We have reconstructed the phylogenetic tree and revised the manuscript accordingly.

We removed one known species (one isolate) namely as “Diaporthe pascoei” from our study as our newly obtained isolate lacks tub gene and the cal and his genes of ex-type are missing. The reconstructed phylogenetic tree showed unsatisfied identification of this species, to avoid further confusion on the species identification we removed it until more data to confirm this. All the related changes have been updated through the whole paper.

Point 1: Complete the Table 1.

Response 1: All newly generated cal and his gene sequences were deposited in the GenBank and access numbers are provided in the Table 1.

Point 2: TEF, TUB, CAL and HIS should written in lowercase and italic.

Response 2: tef, tub, cal and his were written in lowercase and italicized.

Point 3: Line 29 - not clear meaning

Response 3: The phrase was changed by adding the following.

'Newly introduced species in this study could be potentially pathogens'.

Point 4: Line 52

Response 4: The word 'less' was changed to 'few'.

Point 5: Line 166

Response 5: 'Bayesian tree not shown' was removed.

Point 6: Table 2

Point 6) i: Correct the type species

Response 6) i: Type species of the following strains were re-corrected.

Diaporthe citri (CBS 135422)

Diaporthe discoidispora (ZJUD 89)

Diaporthe spinosa (PSCG 383)

Diaporthe subordinaria - No type

Pont 6) ii: Include the his sequences

Response 6) ii: his sequences were included to the following

Diaporthe phragmitis (CBS 138897)

Diaporthe sennicola (CFCC 51634), (CFCC 51635)

Diaporthe sojae (FAU 635)

Point 6) iii: his sequence available in the GenBank for Diaporthe rudis (AR3422)

Response 6) iii: No his sequence is available for Diaporthe rudis (ex-epitype culture AR3422 = CBS 109292). This strain contains only tub, tef, apn2, ITS, act, cal and LSU sequences.

Hence, no his sequence is available in GenBank nor in Udayanga et al. (2014) in the manuscript 'Species limits in Diaporthe: molecular re-assessment of D. citri, D. cytosporella, D. foeniculina and D. rudis. Persoonia 32, 2014: 83–101'.

Point 6) iv: Not D. sojae - BRIP 54033 and CBS 116019

Response 6) iv: Diaporthe kochmanii (BRIP 54033) and Diaporthe phaseolorum (CBS 116019) were synonymed under Diaporthe sojae by Udayanga et al. (2015) in the manuscript 'The Diaporthe sojae species complex: Phylogenetic re-assessment of pathogens associated with soybean, cucurbits and other field crops'. Fungal Biology 119:383–407.

Point 6) v: No type for this species - Diaporthe novem

Response 6) v: According to Santos et al. 2011 (Resolving the Diaporthe species occurring on soybean in Croatia, Persoonia 27, 2011: 9–19), the cultures ex-type is 4-27/3-1 = CBS 127270. Hence we included this strain as the type species in Table 2.

Point 7: Line 339

Response 8: Missing comparison of cal and his were included.

Point 8: Line 344

When you compare D. guttalata with D. angelicae there are not enough differences to be considered different species. These differences may be related to intraspecific variations.

Response 9: Diaporthe guttalata can be distinguished from D. angelicae (7/572 in ITS, 8/467 in tef and 7/453 in tub). However, with the newly addition of cal and his gene regions to the alignment we observed the following base pair changes: 9/606 in cal and 10/513 in his. However, D. cichorii (8/572 in ITS, 13/467 in tef and 7/453 in tub and 21/606 in cal) is the most closely related taxon to D. guttalata as of the Figure 1 and D. cichorii can be clearly distinguished as a separate species by the phylogenetic tree.

Point 9: Line 361

Why you did not add in the tree D. perjuncta? In the ITS tree (on the supplementary material) your isolates group with it, and you have the 5 genes available for D. perjuncta.

Response 10: We have added D. perjuncta to the Table 2 and to the newly constructed phylogenetic analysis (Figure 1). The reference related to this isolate was added to the reference list. The multi-gene (5 genes) phylogenetic analysis indicated that D. irregularis and D. perjuncta are distinct species (Figure 1).

Point 10: Line 438

Compare with D. parenensis, add it on the tree.

Response 11: Diaporthe parenensis has been added to the tree. As of the Figure 1, D. parenensis is phylogenetically distinct from D. minusculata.

Best regards,

Jian-Kui Liu

Round 3

Reviewer 2 Report

  1. The entire manuscript text should be reviewed in relation to English and because it has many confusing parts.
  2. In Material and Methods, lack mention how induced sporulation and what was the subtract used. The subtract should be mentioned also in the photos and on the description of the species.
  3. In table 1, add a column for host.
  4. The discussion needs some more work on it. It is difficult to follow the messages. In the discussion, is missing the description of hosts and worldwide distribution of the species identified in this work.

In the attached pdf you can see more comments and suggestions.

Author Response

Dear Editor and reviewer,

Thank you very much for the review and suggestions made for improving this manuscript. We really appreciate it and have considered all of the major changes. A revised manuscript has been prepared. Responses to reviewer’s comments (point to point) are indicated as required.

The English of whole paper has been checked by Prof. E. B. Gareth Jones (a senior mycologist who has worked in the filed for more than 50 years), and we revised the parts where is necessary.

Point 1: Line 30 - This sentence has to be rewritten, too confusing.

Response 1: The sentence was rewritten as follows.

“Interestingly, the five known Diaporthe species have been reported as pathogens of various hosts, and this could indicate that those newly introduced species in this study could be potentially pathogenic pending further studies to confirm.”

Point 2: Line 46 - None of the references cited uses the 5 genes described. Do change.

Response 2: The following references were added.

38,39,43,56,58,59,75

Point 3: Line 66 - The collected samples were from symptomatic or asymptomatic materials? It should be mentioned.

Response 3: The samples are saprobic. The sentence was revised as follows.

Diaporthe specimens were collected in field surveys of decaying saprobic woody hosts’

Point 4: Line 78 - 2% water agar??

Response 4: Yes it's 2% water agar. This was included to the manuscript.

Point 5: Line 79 - 18 ºC or 25 ºC, which temperature did you used?

Response 5: It's 25 ºC. This was included to the manuscript.

Point 6: Line 84 - missing information about sporulation.

Response 6: We did not include substrates to induce the sporulation.

Point 7:  Line 96 - Besides

Response 7: This word was included.

Point 8: Line 127 - Why did you treated gaps as missing data? gaps and missing data different things!!!

Response 8: We are not able to answer this question which is more likely to be related to the principle of mathematics, and this has been followed with the method of the analysis, in most of the papers in the “M&M” the authors follow this, and We would like to state that this won’t infect the result of our analysis, especially we use the MP/ML/Bayesian as a tool to infer the phylogenetic relationship of our fungi.

Point 9: Line 166 - missing data from "Host"

Response 9: Yes we did not include a column for the hosts, as we collected all samples from unknown decaying wood.

Point 10: Line 200 - how rapid? 2 days..10 days?

Response 10: This was different from one isolate to another. So we provided an overall definition here. However, for each species description, we have provided how many days it takes to cover the entire PDA Petri dishes.

Point 11: Line 202 - after how many days?

Response 11: This fact was also differed from one isolate to another. So we provided an overall definition here.

Point 12: Line 243 - which are immature? and mature? separate them

Response 12: The figures of immature and mature asci were separately annotated.

Point 13: Line 296 - you did not mention the existence of paraphyses but you put a photo of it.

Response 13: Thank you for raising this commen. The following details were included to the species description.

'Paraphyses up to 100 μm long, rarely present, hyaline, smooth, 1–3-septate, cylindrical with obtuse ends, extending above conidiophores'.

Point 14: Line 452 - too confusing please rewrite

Response 14: This was rewritten as follows.

“Morphological characters of the known species isolated in this study were compared with their original descriptions. Phylogenetically, there were no significant base pair differences between these and their type based combined gene alignments.”

Point 15: Line 464 - rewrite. There several studies that demonstrate the usefulness of the 5 selected genes

Response 15: This was rewritten as follows.

‘Dissanayake et al. [10] reviewed the genus Diaporthe and provided a checklist for 171 species with available molecular data (from culture and fruiting body) and a phylogenetic tree using four gene regions (ITS, tef, tub and cal). According to Santos et al. [56], incorporation of a five-loci dataset (ITS, cal, his, tef, tub) was recommended as the best combination for species identification within the genus and the recent studies seems to stick to the selection of 4 or 5 genes [33–43]. Hence, the present study is conducted combining the five gene regions analyses of ITS, tef, tub, cal and his to reveal five known Diaporthe species and to assist in the introduction of seven new Diaporthe species’.

Point 16: Line 470 - rewrite, hard to follow

Response 16: This was rewritten as follows.

‘Several studies have been conducted to reveal the association of Diaporthe species with various hosts in China. Huang et al. [66] revealed seven apparently undescribed endophytic Diaporthe species (Diaporthe biconispora, D. biguttulata, D. discoidispora, D. multigutullata, D. ovalispora, D. subclavata and D. unshiuensis) on Citrus. Gao et al. [67] identified four novel species (D. apiculata, D. compacta, D. oraccinii, D. penetriteum) and three known species (D. discoidispora, D. hongkongensis, D. ueckerae) associated with Camellia (tea). Gao et al. [68] showed eight new species of Diaporthe (Diaporthe acutispora, D. elaeagni-glabrae, D. incompleta, D. podocarpi-macrophylli, D. undulata, D. velutina, D. xishuangbanica and D. yunnanensis) from leaves of several hosts while Yang et al. [37] introduced twelve new Diaporthe species (Diaporthe acerigena, D. alangii, D. betulina, D. caryae, D. cercidis, D. chensiensis, D. cinnamomi, D. conica, D. fraxinicola, D. kadsurae, D. padina and D. ukurunduensis) from infected forest trees in Beijing, Heilongjiang, Jiangsu, Jiangxi, Shaanxi and Zhejiang Provinces. Three new Diaporthe species: Diaporthe anhuiensis, D. huangshanensis, D. shennongjiaensis and two other known species: D. citrichinensis and D. eres were described as endophytes by Zhou et al. [42]. Yang et al. [39] established three new species: D. albosinensis, D. coryli and D. shaanxiensis isolated from symptomatic twigs and branches at the Huoditang Forest Farm in Shaanxi Province, China. High diversity of Diaporthe species associated with pear shoot canker in China was observed by Guo et al. [43] representing thirteen known species (Diaporthe caryae, D. cercidis, D. citrichinensis, D. eres, D. fusicola, D. ganjae, D. hongkongensis, D. padina, D. pescicola, D. sojae, D. taoicola, D. unshiuensis and D. velutina) and six new species (Diaporthe acuta, D. chongqingensis, D. fulvicolor, D. parvae, D. spinosa and D. zaobaisu). However, the identification of Diaporthe species associated with hosts in nature reserves in China has rarely been studied. Thus an investigation of Diaporthe species was conducted and this provides the first molecular phylogenetic frame of Diaporthe diversity in six nature reserves in Karst region of Guizhou province, combined with morphological descriptions’.

Point 17: Line 498 - Delete

Response 17: This sentence was removed from the manuscript.

Thanks again for your effort on our paper.

Best regards,

Jian-Kui Liu